# Low-input PacBio sequencing generates high-quality individual fly genomes and characterizes mutational processes

Hangxing Jia [1,5] ✉, Shengjun Tan[1,5] ✉, Yingao Cai [1,2,5], Yanyan Guo[1,2], Jieyu Shen [1,2], Yaqiong Zhang [1], Huijing Ma[1], Qingzhu Zhang[1,2], Jinfeng Chen [2,3], Gexia Qiao[1,2], Jue Ruan [4] ✉ & Yong E. Zhang [1,2] ✉

Long-read sequencing, exemplified by PacBio, revolutionizes genomics, overcoming challenges like repetitive sequences. However, the high DNA requirement (>1 μg) is prohibitive for small organisms. We develop a low-input (100 ng), low-cost, and amplification-free library-generation method for PacBio sequencing (LILAP) using Tn5-based tagmentation and DNA circularization within one tube. We test LILAP with two *Drosophila melanogaster* individuals, and generate near-complete genomes, surpassing preexisting single-fly genomes. By analyzing variations in these two genomes, we characterize mutational processes: complex transpositions (transposon insertions together with extra duplications and/or deletions) prefer regions characterized by non-B DNA structures, and gene conversion of transposons occurs on both DNA and RNA levels. Concurrently, we generate two complete assemblies for the endosymbiotic bacterium *Wolbachia* in these flies and similarly detect transposon conversion. Thus, LILAP promises a broad PacBio sequencing adoption for not only mutational studies of flies and their symbionts but also explorations of other small organisms or precious samples.

Evolution has sculpted extraordinary biodiversity, with each species enriching our lives or shedding light on problems in medicine, agriculture, conservation, and bioindustry[1]. The genome is central to investigating a species, and genome sequencing and assembly have been revolutionized by Pacific Biosciences (PacBio) High Fidelity (HiFi) and Oxford Nanopore Technology (ONT) long-read sequencing technologies[2–4]. Due to the prerequisite of a high DNA input (>1 μg)[5], long-read sequencing is suitable for large organisms or studies with abundant DNA. However, for one predominant lineage of biodiversity, insects[6–8], a considerable proportion of species are diminutive. The insect model, *Drosophila melanogaster*, yields only ~100–150 ng of DNA from a male individual[9,10]. Many small insects cannot be raised

and inbred in the lab or show high heterozygosity, even including cryptic species, making high-quality assemblies impossible to generate through population sequencing[11–13]. This predicament is reflected in the less than 1% of insects with reported genomes[7,14]. The DNA input requirement is well known to be a major challenge in megaprojects such as the Earth BioGenome Project, which aims to sequence all eukaryotic genomes[1,15]. Thus, there is a pressing need for a low DNA input, long-read sequencing method, enabling genome assembly of a single insect.

Two strategies have emerged to reduce DNA input, but at the cost of time, financial resources, or sequencing completeness (Fig. 1a). Specifically, the PacBio company developed a ligation-based low-input

[1]Key Laboratory of Zoological Systematics and Evolution, Institute of Zoology, Chinese Academy of Sciences, Beijing, China. [2]University of Chinese Academy of Sciences, Beijing, China. [3]State Key Laboratory of Integrated Management of Pest Insects and Rodents, Institute of Zoology, Chinese Academy of Sciences, Beijing, China. [4]Shenzhen Branch, Guangdong Laboratory of Lingnan Modern Agriculture, Genome Analysis Laboratory of the Ministry of Agriculture and Rural Affairs, Agricultural Genomics Institute at Shenzhen, Chinese Academy of Agricultural Sciences, Shenzhen, China. [5]These authors contributed equally: Hangxing Jia, Shengjun Tan, Yingao Cai. ✉e-mail: jiahangxing@ioz.ac.cn; tanshengjun@ioz.ac.cn; ruanjue@caas.cn; zhangyong@ioz.ac.cn

**a**

|  | DNA input | Cost per library | Time cost per library | Reagent dependency | Equipment dependency | Amplification bias and chimerism |
|---|---|---|---|---|---|---|
| PacBio low-input method | 400 ng | High (~100 dollars) | >3.5 hours | PB Template Prep Kit, Ampure PB beads | Megaruptor System, Femto Pulse System | No |
| Amplification-based methods | 1–10 ng or single cells | High (>100 dollars) | Hours to 1 day | | | Yes |
| LILAP | 100 ng | Low (~10 dollars) | 2 hours | No | No | No |

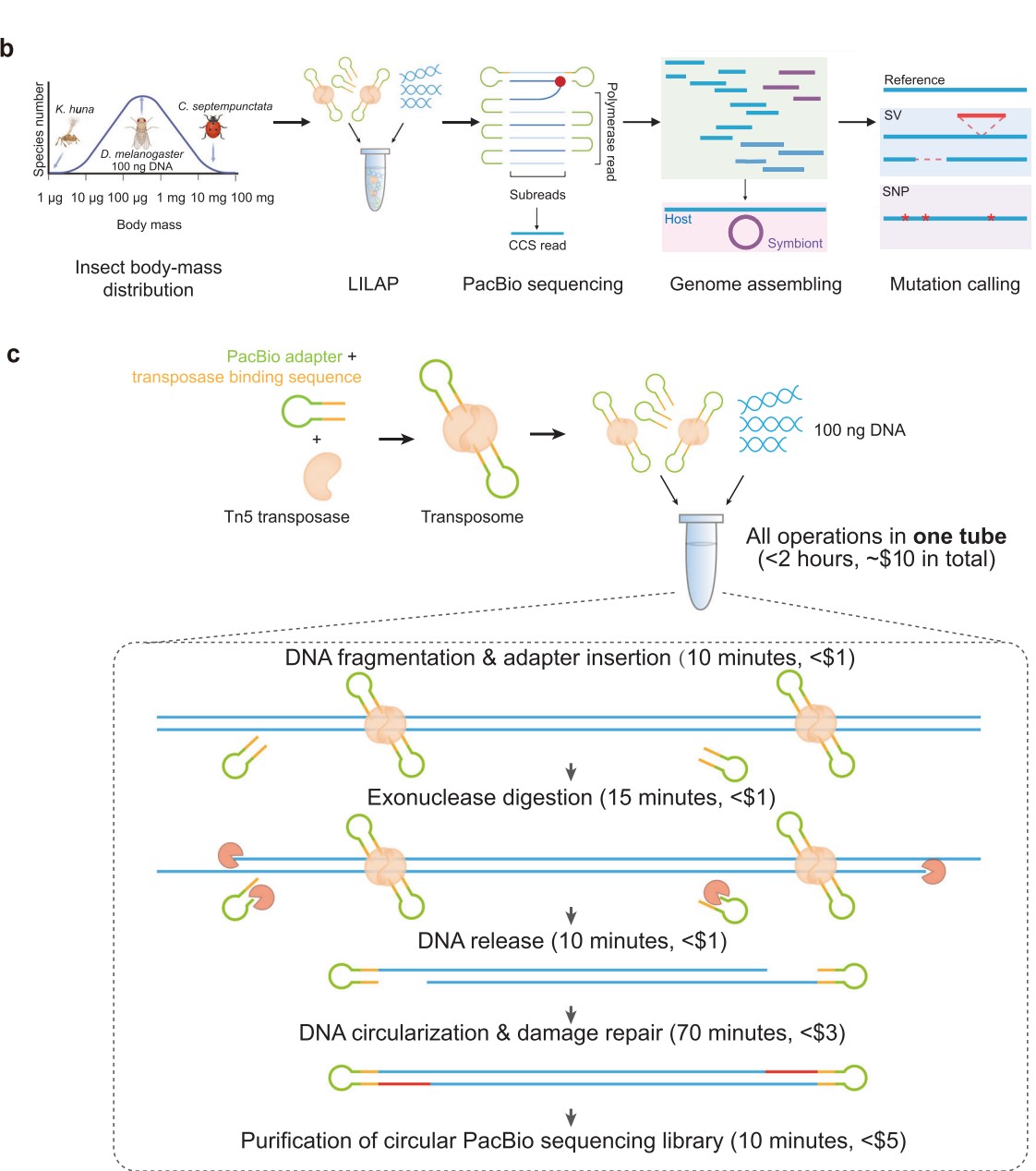

method of library preparation and reduced DNA input to 100 ng for the Sequel system and 400 ng for the Sequel II system[16,17]. Subsequent advancement further lowered the DNA input requirement to 1–10 ng or even to single cells for PacBio or ONT sequencing by integrating DNA amplification and standard library preparation[9,12,18–20]. Regrettably, both PacBio low-input and amplification-based methods entail substantial time and financial investments, necessitating specialized reagents and equipment (Fig. 1a). This is primarily due to the inherent intricacies of these protocols (Supplementary Fig. 1). Moreover, amplification-based methods suffer from both low genomic coverage (caused by amplification bias) and chimerism[21,22]. These problems are exemplified by previous studies involving single-fly genomes,

**Fig. 1 | Summary of low-input library preparation methods, project design, and schema of LILAP. a** Comparison of low-input methods on the PacBio sequencing platform. "DNA input" pertains to the currently commonly employed PacBio Sequel II platforms. PB Template prep kit is used to prepare the standard PacBio library, AMPure PB beads are used for DNA size selection and DNA recycling, Megaruptor is used for DNA fragmentation, and Femto Pulse System is used for size selection. **b** The overall design of this study. A log-scaled axis was used for the leftmost panel. Cartoons for three exemplar insects are shown with *Kikiki huna* in the left, *D. melanogaster* in the middle, and *Coccinella septempunctata* in the right. For PacBio sequencing, the red point indicates polymerase and the yellow line marks the transposase binding sequence (see also Panel **c**). CCS: circular consensus sequencing; SV: structural variation; SNP: single nucleotide polymorphism. **c** Flowchart of LILAP. Two Tn5 molecules bind to two hairpin adapters to form an active homodimer transposome. After tagmentation by mixing the gDNA and transposome, the hairpin adapters are tagged to gDNA, and then the DNA fragments are circularized by gap filling and nick ligation. Panel b was created with BioRender.com released under a Creative Commons Attribution-NonCommercial-NoDerivs 4.0 International license.

where the de novo-assembled genomes cover only 75–85% of the reference genome[9,10].

We herein developed a straightforward, low-input (100 ng), low-cost, and amplification-free library-generation method for PacBio sequencing (LILAP) and tested it in *D. melanogaster* individuals. LILAP minimizes DNA loss by performing Tn5-based tagmentation, DNA circularization, and damage repair within one tube. We evaluated LILAP in two individuals of the ISO1 reference strain based on three considerations: (1) the two aforementioned works tested their methods in flies[9,10], making method comparison feasible; (2) male flies, given their proximity to the median body mass for insects[23,24], representing this lineage (Fig. 1b); and (3) mutations carried by individual ISO1 flies tend to be recently generated, providing a more accurate reflection of mutational properties compared to relatively older mutations found in a pool of individuals or in non-ISO1 flies, which are likely confounded by secondary mutations[25,26]. We therefore de novo assembled two individual genomes, resulting in substantially improved quality compared to previous individual-based assemblies. To further demonstrate the value of LILAP, we analyzed these two high-quality genomes and identified two mutational properties in which non-B DNA structures are associated with complex transposition events [transposable element (TE) insertions with extra duplications and/or deletions] and DNA- or RNA-level gene conversion events homogenize TEs. In parallel, we assembled two genomes of endosymbiotic *Wolbachia* and similarly identified TE conversion. Altogether, our evaluations and analyses highlight the power of LILAP in resolving the genomes of small organisms together with their endosymbionts and provide insights into mutational processes. This method thus paves the way for the large-scale sequencing of insects or other small organisms.

## Results

### LILAP is streamlined with all reactions performed within one tube

To generate circular PacBio sequencing libraries, LILAP requires five steps (Fig. 1c, "Methods"). First, Tn5 transposase mediates tagmentation, where adapter ligation and genomic DNA (gDNA) fragmentation are achieved simultaneously[27,28]. In contrast to linear adapters that are widely used in short-read library building[28–30], we designed a hairpin adapter consisting of the PacBio sequencing adapter and the 19-bp mosaic end (ME) sequence recognized by Tn5 (Fig. 1c). Optional barcodes could be added in the middle of the sequencing adapter and ME. By mixing the transposome (hairpins bound by Tn5 homodimers) and gDNA, tagmentation occurs where adapters are inserted, and gDNA is fragmented with 9-bp gaps at the 3' end of each strand. Second, the extra adapters and DNA ends unprotected by Tn5 transposase are removed by exonucleases. Third, exonucleases and Tn5 transposase are inactivated, and the DNA fragments previously bound by Tn5 transposase are released. Fourth, these fragments are circularized via gap filling and nick ligation, while possible DNA damage generated in the library-building process is repaired. The fifth and final purification step prepares the DNA library for sequencing. Note that LILAP is robust, as demonstrated by a similar amount of circular libraries across four biological replicates (Supplementary Fig. 2a).

In these five steps, LILAP implements three essential strategies, including tagmentation, digestion of extra adapters, and one-tube

library building (Fig. 1c, "Methods"). Tagmentation is accomplished through Tn5-mediated simultaneous fragmentation and adapter tagging, an efficient technique that is widely used in short-read sequencing library building[28,31–33]. Notably, extra adapters compete with gDNA fragments for DNA ligase, which can impair DNA circularization. Exonuclease digestion is thus crucial for efficient circularization due to the presence of abundant Tn5-unbound adapters in standard tagmentation experiments[34]. The one-tube pipeline strategy minimizes gDNA loss caused by recurrent DNA recycling in standard library preparations. To enable sequential reactions within one tube, a universal buffer is critical to ensure that the later reactions are not disturbed by reagents in the early reactions. We determined that T4 DNA ligase buffer was suitable for the following reasons: (1) T4 DNA ligase buffer meets the requirements for T4 DNA ligase (used for circularization, "Methods"); and (2) among three commercially available Tn5 transposases, both Vazyme Tn5 and M5 Tn5 showed high efficiency in T4 DNA ligase buffer (Supplementary Fig. 2b), and we chose the latter due to its low price. Referring to previous strategies for enzyme inactivation[18,30], we added SDS to inactivate both the Tn5 transposase and exonucleases to release gDNA (Supplementary Fig. 2c). We then added Triton X-100 to quench SDS and minimize its influence on subsequent reactions.

With the straightforward design, LILAP requires less than 2 h, costs about $10 in reagents, and operates independently of specialized instruments (Fig. 1c). In contrast, prior PacBio low-input or amplification-based methods exhibit longer time, elevated costs, and instrument dependency (Fig. 1a).

### LILAP-based high-quality sequencing of two individual flies

We tested LILAP in two male adults of the ISO1 strain (hereafter referred to as ISO1-1 and ISO1-2, respectively) using 100 ng gDNA extracted from each individual. Note that the PacBio HiFi platform generated high-quality circular consensus sequencing (CCS) reads based on polymerase reads consisting of multiple copies of error-prone subreads (Fig. 1b). Thus, to increase the quality of CCS reads with a low DNA input, we optimized library insert size based on the following three facts: (1) the quality of CCS reads increases with an increase in the copy number of subreads or passes and plateaus in 15–20 passes[35]; (2) a shorter insert size or subread length contributes to a higher DNA library concentration; and (3) the median length of polymerase reads is ~100 kb[35]. Thus, we generated two libraries with a median read length of ~4.5 kb for CCS reads (Fig. 2a, "Methods"). That is, although we extracted gDNA longer than 20 kb (Supplementary Fig. 2d, "Methods"), we followed the routine practice[20,36] of adjusting the relative quantities of Tn5 and gDNA to construct shorter libraries. With one sequencing cell of the PacBio Sequel II platform used for either ISO1-1 or ISO1-2, 7.7–8 gigabytes (GB) of CCS reads were generated. Consistent with a previous report[35], the sequencing accuracy plateaued after 15–20 passes. The overall median accuracy of CCS reads reached QV36 (Phred-scaled Quality Value 36), which is equivalent to a base error rate of 0.02% (Fig. 2b).

In addition to high accuracy, CCS reads show high chromosomal uniformity and low GC bias. Specifically, sex chromosomes show the expected relative depth (i.e., half that of autosomes, Fig. 2c). More

 

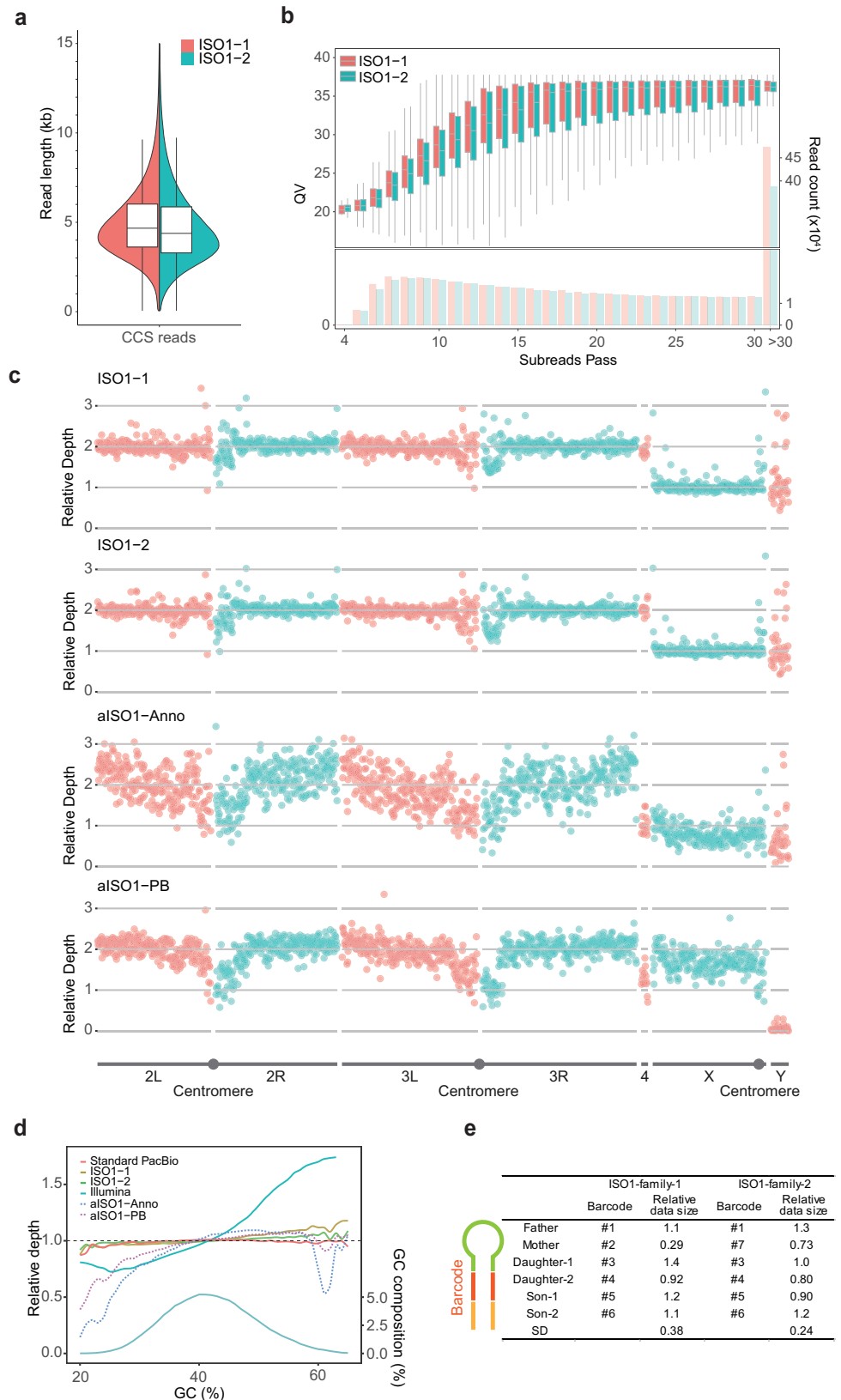

importantly, coverage depths are largely consistent for most regions except for centromeres and chrY, which have high repeat contents[35,37], potentially disturbing read mapping. For GC bias, since Tn5 transposase tends to tag high-GC regions in short-read sequencing library preparation[36], we examined how severe this effect was in LILAP. As expected, short-read sequencing data showed a significant excess

(>50%) in high-GC regions (Fig. 2d). In contrast, ISO1-1/2 displayed a low excess (<20%) in high-GC regions compared to standard PacBio sequencing data with a large DNA input (>1 μg)[38]. This difference could be attributed to PCR amplification during the construction of short-read sequencing libraries, which may exacerbate GC bias, in addition to Tn5 binding[39].

**Fig. 2 | Evaluation of LILAP sequencing data. a** The length distribution of CCS reads. The curves show the probability density (ISO1-1: $n = 1723578$, ISO1-2: $n = 1722440$). **b** Box plot showing the relationship between the QV of each CCS read and its subread passes. Bar chart indicating read count in each subreads pass. Reads with pass numbers higher than 30 were merged into the last bin. The left and right $Y$-axes represent QV and read count, respectively. Only reads with lengths between 3500 bp and 6000 bp were displayed (ISO1-1: $n = 838596$, ISO1-2: $n = 744864$). **c** The chromosomal distribution of relative read depth with 100-kb window size. Annoroad's amplification-based sequencing data were denoted as aISO1-Anno, whereas PacBio's data were labeled as aISO1-PB. Except for the aISO1-PB, all other datasets were derived from male flies. Consequently, the presence of a minority of reads mapped to chrY in aISO1-PB likely indicates mismapping. Notably, the dots positioned between chr2L (Left) and chr2R (Right), between chr3L and chr3R, and within chrX denote centromeres. **d** The relative read depth in regions with different GC content. Standard PacBio HiFi sequencing data, Tn5 tagmentation-based short-read (Illumina) data, and amplification-based PacBio HiFi sequencing data served as controls. The left and right $Y$-axes show the relative depth and the proportion of the corresponding GC bins, respectively. The relative depth was calculated as $\frac{depth\ of\ each\ window}{median(depth\ of\ all\ windows)}$. **e** The relative data yield of two multiplexed HiFi sequencing libraries. Each contains a fly family. The relative data size was calculated as $(\frac{the\ nucleotide\ number\ of\ CCS\ reads\ from\ each\ single\ fly}{the\ total\ nucleotide\ number\ of\ CCS\ reads\ from\ six\ flies}) \times 6$. SD stands for standard deviation. For Panels (**a**, **b**), boxes represent the first and third quartiles, with medians marked with the middle line, while whiskers represent the minimum and maximum values.

The remarkable coverage evenness likely stems from the amplification-free nature of LILAP or the absence of amplification bias[21]. To investigate this, we compiled two sets of individual fly sequencing data using standard amplification protocols on the PacBio Sequel II platform: one produced by Annoroad's commercial service (referred to as aISO1-Anno), and the other by PacBio itself (referred to as aISO1-PB, "Methods"). As anticipated, both aISO1-Anno and aISO1-PB show more pronounced fluctuations (Fig. 2c), with low- or high-GC regions being underrepresented, displaying depths 10~80% lower than expected (Fig. 2d).

We then analyzed the data yield and data uniformity across barcoded samples by generating multiplexed sequencing data. For a PacBio HiFi library covering an entire fly family (the parents and four offspring), the standard deviation (SD) of the data yield was 0.38 (Fig. 2e and Supplementary Data 1). This moderate variance was mainly contributed by the underrepresented "mother" sample, which was attached to barcode #2 (Supplementary Data 2). Therefore, we generated another multiplexed library by replacing #2 with a distinct barcode, #7, and obtained a more even sequencing depth (SD = 0.24).

In summary, LILAP produces high-quality sequencing data in terms of read accuracy, uniformity across chromosomes, GC content, and barcode dimensions.

## De novo assembly of LILAP data generates two high-quality individual fly genomes

By integrating de novo assembly pipelines in both insects and humans[40,41], we designed a computational method for assembling and polishing individual fly genomes based on LILAP data ("Methods"). This pipeline could be roughly divided into two steps: the use of hifiasm[42] as the assembler, given its widespread usage and strong performance[12,40], and the further polishing of the initial assemblies by identifying and correcting homozygous mutations (potential assembly errors)[41]. Similarly, we conducted assemblies for two individual fly genomes using standard amplification protocols: aISO1-Anno and aISO1-PB. Subsequently, we assessed our final ISO1-1/2 assemblies against the aISO1-Anno/PB assemblies, alongside five previously published long-read fly assemblies. These include two individual fly assemblies[9,10] and three assemblies generated by sequencing dozens or hundreds of flies[38,43,44]. This ensemble of nine assemblies encompasses variations in sequencing platforms and depths, thereby providing a comprehensive control for evaluation.

Overall, the ISO1-1/2 assemblies are on par with the previous best assembly generated by HiFi sequencing of pooled flies in terms of base accuracy and gene/genome completeness, while surpassing other assemblies (Table 1). Specifically, the genome-wide QV of the initial assemblies generated by hifiasm reached ~60 ($10^{-6}$ error rate) and increased to 63~65 after polishing. In contrast, QVs for two earlier PacBio-based assemblies ranged from 42 (generated in the less accurate continuous long-read mode or CLR mode) to 63 (HiFi mode), while the three ONT-based assemblies exhibited lower values (28~38). This discrepancy aligns with the accuracy gradient across HiFi, CLR, and ONT sequencing[45,46]. Moreover, 98.7% of the conserved single-copy genes were assembled in the ISO1-1/2 assemblies, while the proportions for previous assemblies were generally lower (median: 97.2%). The reference genome coverage exhibited even more pronounced disparities, with the ISO1-1/2 assemblies achieving 99.6% coverage for all major chromosomes, excluding the poorly assembled chrY[47–49], whereas coverage for other assemblies varied between 80.9% and 99.2%. Notably, all three ONT-based assemblies showed low genomic coverage (80.9–91.7%), potentially attributable to their lower base accuracy, leading to challenges in resolving highly similar repeats. The larger assembly sizes (164–167 Mb) of ISO1-1/2 further signify enhanced completeness. While the total genome size of a male fly approximates 180 Mb[36,46], the reference genome size is only 144 Mb primarily due to difficulties in assembling the repeat-rich chrY (~40 Mb[47,48]). Two previous PacBio-based assemblies also showed different sizes for samples of different sexes (164 Mb for males and 147 Mb for females; Table 1).

The inclusion of two amplification-based assemblies (aISO1-Anno/PB) in our evaluation underscores the advantage of amplification-free practices. Despite utilizing the same sequencing platform (Sequel II HiFi) and similar sequencing depth (60x), both aISO1-Anno and aISO1-PB demonstrate comparatively lower genome coverage (98.2 and 97.9%, Table 1), potentially attributable to amplification bias (Fig. 2d). Consistent with this observation, they also exhibit relatively lower QV (55 and 56), likely stemming from amplification errors caused by PCR[50]. Even with an increase in sequencing depth (from 60x to ~185x), these two metrics for both amplification-based assemblies exhibit only moderate changes (Supplementary Data 3), indicating that a sequencing depth of 60x suffices and that amplification bias or errors are not effectively mitigated by further depth augmentation.

Genome-wide parameters such as coverage and size represent a simplified gross status. To further evaluate the completeness of ISO1-1/2, we examined whether a notorious example, the *Sdic* tandem gene family locus[51], could be resolved. This locus is difficult to assemble due to high sequence similarity between paralogous genes or paralogous intergenic regions, with similarities reaching up to 96% and 99%, respectively (Supplementary Fig. 3a). These complexities, combined with sequencing errors, have led to conflicting reports, showing four, seven, or six copies, among which six copies have been proven to be correct[52]. With our CCS reads averaging ~4.5 kb in length (Fig. 2a) and exhibiting a low sequencing error rate (0.02%, Fig. 2b), each read is anticipated to contain an ample number of authentic sites capable of distinguishing between paralogous regions. Coupled with adequate sequencing depth (Supplementary Fig. 3b), we achieved the successful assembly of a single contig in ISO1-1/2 assemblies, with all six copies arranged in the correct order (Supplementary Data 4).

The only disadvantage of LILAP is the short continuity as quantified by contig NG50 (the length at which half of the predicted genome size is contained in contigs greater than this length)[53]. This results from the lack of an independent scaffolding method (e.g., Hi-C[9]) and a short insert size (4.5 kb vs. 8.2 kb or higher, Fig. 2a and Supplementary Fig. 4). However, even for this parameter, ISO1-1/2 showed a value of 6.7 Mb (Table 1), which is better than aISO1-Anno (1.2 Mb) and the two

**Table 1 | Evaluation of ISO1-1/2 assemblies and other long-read fly assemblies**

| Study[a] | Platform[b] | Sequencing depth | DNA input[c] | Sex[d] | QV[e] | Genome coverage[f] | Gene coverage[g] | Size (Mb) | NG50 (Mb) | Median LG90[h] |
|---|---|---|---|---|---|---|---|---|---|---|
| ISO1-1 | PacBio Sequel II HiFi mode | 59x | 100 ng | M | 60.6/64.6 | 99.6% | 98.7% | 164.1 | 6.7 | 5 |
| ISO1-2 | | 57x | | M | 58.5/63.1 | 99.6% | 98.7% | 167.5 | 6.7 | 4 |
| aISO1-Anno | PacBio Sequel II HiFi mode | 60x | 40 ng | M | 54.7 | 98.2% | 97.9% | 156.5 | 1.2 | 20 |
| aISO1-PB | | 60x | 10 ng | F | 56.3 | 97.9% | 98.7% | 156.4 | 9.7 | 4 |
| Berlin et al.[43] | PacBio Sequel II CLR mode | 90x | 10 μg | M | 42.9 | 99.2% | 98.3% | 164.1 | 13.6 | 2 |
| Adams et al.[9] | ONT+ Illumina+Hi-C | 60x | 78 ng | F | 33.6 | 80.9% | 92.6% | 111.0 | 21.7 | NA |
| Heavens et al.[10] | ONT R9.4 | 36x | 110 ng | Un | 37.7 | 91.7% | 94.4% | 129.8 | 0.6 | 86 |
| Solares et al.[44] | ONT R9.5 | 30x | 1.5 μg | Mixed | 28.5 | 82.8% | 97.2% | 140.7 | 2.9 | 11 |
| Nurk et al.[38] | PacBio Sequel II HiFi mode | 40x | >1 μg | F | 63.3 | 97.2% | 98.8% | 147.3 | 20.2 | 1 |

[a]Adams et al. used a hybrid individual derived from the ISO1 and I38 strains[9], while aISO1-PB and Nurk et al. used a hybrid population of ISO1 and A4[38]. Other studies used the ISO1 strain exclusively. The assembled genome from Heavens et al. was not publicly available[10], and we performed assembly independently ("Methods"). In contrast, the other four studies made their assemblies public, and the statistics pertain to those assemblies.

[b]Adams et al. generated multiple sequencing data types from both the ONT and Illumina platforms[9] and sequencing depth and DNA input values refer to the ONT data. Note that the ONT Flow Cell information is not available.

[c]The study by Nurk et al. lacks specific DNA quantity information[38]. Nevertheless, considering the use of pooled flies, it is reasonable to infer that the DNA input exceeded 1 μg. DNA input for Adams et al. specifically refers to the ONT sequencing.

[d]"M", "F" and "Un" refer to male, female, and unknown sexes, for which Heavens et al. did not provide specific information[10].

[e]For ISO1-1/2 assemblies, QV values for both the initial and polished assemblies are presented, while for other assemblies, no further polishing was performed.

[f]To account for chromosomal differences between sexes, coverage was calculated across all chromosomes except chrY.

[g]Gene coverage was quantified using BUSCO complete gene models ("Methods").

[h]"Median LG90" was exclusively computed for chr2L, chr2R, chr3L, chr3R, and chrX. "NA" denotes the inability of the assembly from Adams et al. to cover 90% of the reference genome across the five arms.

ONT-based assemblies (0.6–2.9 Mb). Consistently, their median LG90s (number of contigs covering over 90% chromosomes) are only 4 or 5 (Table 1 and Supplementary Fig. 5), showcasing a reasonable level of continuity.

Collectively, the results indicated that the ISO1-1/2 assemblies largely outperformed other long-read assemblies from single or pooled flies, achieving a base error of $10^{-6}$ and near-complete coverage of reference genomes or genes.

## Error-prone DNA repair mechanisms are involved in transposition and duplication

In the two high-quality individual genomes, we analyzed mutations, especially those types that have posed challenges in short-read data, such as structural variations (SVs)[54–56]. Despite the global collinearity between ISO1-1/2 and the reference genome (Supplementary Fig. 6), we identified 337 SVs in ISO1-1/2 (Fig. 3a). All these SVs involved structural alterations of known sequences present in the reference genome, such as TE insertions or duplications. Two lines of analysis indicated the high quality of this dataset ("Methods"). First, since these two flies were randomly sampled from one maintenance tube and were highly related, most SVs were shared (318/337 or 94.4%, Fig. 3a). Shared SVs, which likely accumulated in the lab strain's ancestors, are expected to be homozygous in one individual and to be present in the two aforementioned families (Fig. 2e). Indeed, consistent patterns were revealed (Fig. 3b). To eliminate the likelihood that observed SVs are artifacts from Tn5-mediated library preparation, we manually assessed 80 randomly selected shared SVs for their presence in aISO1-Anno, which was derived from the identical laboratory strain. Consistently, most SVs (78 or 97.5%) were confirmed present, while the remaining two could be explained by between-individual differences (Supplementary Data 5). Second, by exploiting one previous TE insertion dataset[25], we found that ISO1-1/2 shared 63 TE insertions with another ISO1 substrain (ISO1-BL) but only 1 ~ 2 events with three non-ISO1 strains (Supplementary Fig. 7a). Moreover, in comparison to the reference genome, ISO1-1/2 or ISO1-BL carried approximately 200 TE insertions, whereas each of the remaining strains possessed ~500 or more insertions. The latter pattern gets more pronounced when considering all types of SVs in the published genomes of 13 *Drosophila* synthetic population resources (DSPR) founder strains (median: 3,241, Supplementary Fig. 7b)[26]. The substantial evolutionary divergence between DSPR strains and ISO1 accounts for 12.8% of SVs, which were complicated by secondary mutations[26].

A reliable SV dataset, coupled with the small evolutionary distance between ISO1-1/2 and the ISO1 reference genome, largely mitigating the confounding influence of secondary mutations in distantly related non-ISO1 strains, facilitates the subsequent analysis of mutation properties. We noticed that the SV dataset primarily comprised TE insertions, among which those in ISO1-1/2 and the reference genome accounted for 56.1% (189) and 22.3% (75), respectively (Fig. 3a). TE types diverged between two categories: DNA TEs, particularly *hobo* elements (*hAT* superfamily members), abounded in ISO1-1/2 (Fig. 3c), while LTR retrotransposons, especially *Gypsy* elements, dominated the reference genome (Supplementary Fig. 7c). These patterns are consistent with previous findings, indicating that LTR retrotransposons are the most active TE subclass in the reference genome[57], while *hobo* elements frequently show high activity in laboratory strains[25,58,59].

We next examined the presumably younger TE insertions (the 189 insertions in ISO1-1/2) and unexpectedly identified nine complex transposition events, characterized by TE insertions with extra duplications and/or deletions. Notably, all nine cases were once again confirmed in aISO1-Anno (Supplementary Data 5). Six cases involved *hobo* insertions, and three involved the non-LTR retrotransposon *Jockey* (Fig. 3d and Supplementary Fig. 8). For eight out of nine cases, deletions ranging from 3 to 380 bp were present in the immediate flanking region (median: 190 bp; Fig. 3e and Supplementary Fig. 8). Two *hobo* cases were linked to duplications: (1) at chr2L: 9.4 Mb, a proximate 10-bp

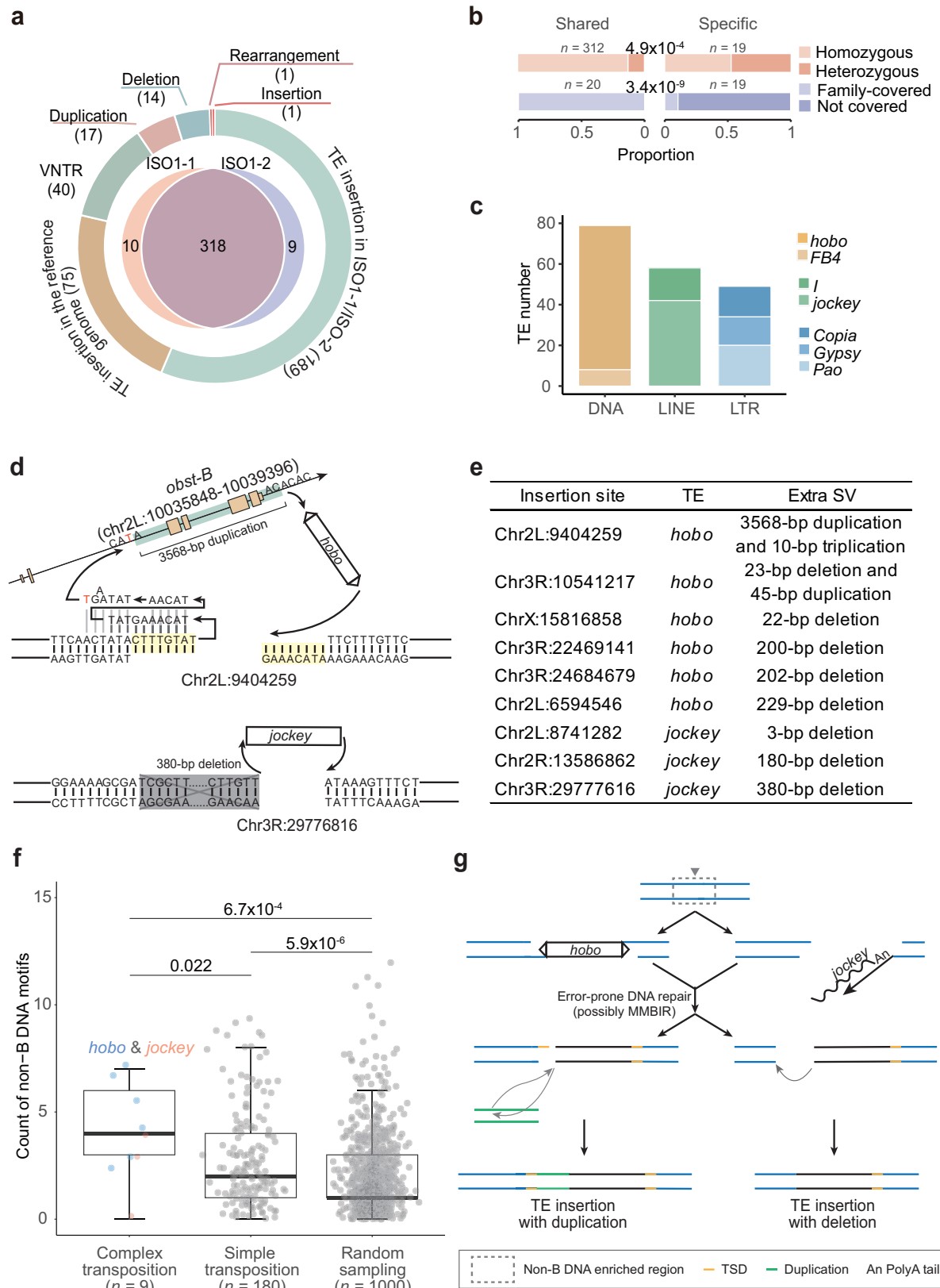

sequence triplication occurred together with a duplication of the 3568-bp *obst-B* segment at chr2L: 10.0 Mb (Fig. 3d); (2) at chr3R: 10.5 Mb, a proximate 23-bp deletion occurred alongside a duplication of the 45-bp sequence at chr2L: 12.1 Mb (Supplementary Fig. 8).

Four findings suggested that these complex cases result from transposition followed by error-prone DNA repair, inducing extra

SVs. First, the terminal inverted repeats (TIRs) of six *hobo* elements remained intact, and three *Jockey* elements were almost entirely preserved, showing 5' truncation of several bases. Second, the hallmark sequence flanking transposons [i.e., target site duplication (TSD) caused by the repair of two staggered DNA nicks after transposon integration[60,61]] was present in the aforementioned case at

**Fig. 3 | Structural variations in two single male flies. a** Counts of various types of SVs in ISO1-1/2. VNTR indicates the variable number of tandem repeats. **b** The proportion of shared or individual-specific SVs in terms of homozygosity and occurrence in fly families. **c** The distribution of transposons in ISO1-1/2. For DNA and LINE transposons, individual element names are directly displayed, as only one element was identified. In the case of LTR retrotransposons, names of superfamilies are provided, as multiple elements were identified. **d** Two complex transposition examples: *hobo* insertion associated with a short triplication involving TSD on one side and a duplication of *obst-B* (top); and *Jockey* insertion associated with a deletion (bottom). Arrows indicate sequence arrangement order. In the top example, a one-bp microhomology ("T") is marked in red, while the TSD or *hobo* recognition motif appears in yellow. For *obst-B*, the green box includes the sequences duplicated from the *obst-B* locus, larger orange boxes represent coding exons, smaller ones denote UTRs, and connecting lines indicate introns. Triplication is error-prone when "GA" is missing and an extra "A" is inserted. In the bottom example, the deletion is marked in grey. **e** A list of nine complex transpositions. **f** Distribution of non-B DNA motif counts on TE insertion sites (up- and downstream 200-bp window) and random 400-bp windows. Complex transposition represents TE insertion with extra duplications and/or deletions. The bar indicates the interquartile range, the whiskers indicate minima and maximums, and the thick line indicates the median. **g** A schematic model to explain the formation of complex transpositions through error-prone DNA repair.

chr2L:9.4 Mb but absent from the other eight cases due to immediate flanking deletions (Fig. 3d and Supplementary Fig. 8). Third, breakpoints in all *hobo* cases fit the known 8-bp recognition motif of *hobo* transposase (5′-nTnnnnAn-3′; Fig. 3d and Supplementary Fig. 8)[61,62]. Fourth, two *hobo* cases involving duplications fit the microhomology-mediated break-induced replication (MMBIR) model[63], where complex rearrangements are induced by replication slippage and template switching facilitated by microhomology (a short similar sequence at the breakpoint; e.g., "T" in *obst-B*, Fig. 3d). MMBIR can also generate deletions, as observed in eight cases (Fig. 3e), although the possible involvement of other repair mechanisms (e.g., nonhomologous end joining, NHEJ) cannot be excluded, particularly as they mainly generate deletions[64–66].

Since double-strand breaks (DSBs) of non-B DNA structures are often associated with error-prone repair pathways (e.g., MMBIR[67,68]), we examined whether nine complex transpositions occurred at loci with a larger number of non-B DNA motifs than loci with simple transpositions ("Methods"). Indeed, these nine loci harbored significantly more non-B motifs than loci with simple transpositions or randomly sampled loci (median: 4, 2, and 1; Fig. 3f). The contrast between the latter two categories is also significant, suggesting that transpositions generally prefer non-B DNA regions. These patterns remained robust with a larger window size (Supplementary Fig. 9).

All of these data suggest the following model of complex transposition (Fig. 3g): (1) for either *hobo* or *jockey* elements, their sequences have been integrated, and one nick has been repaired as in simple transposition; (2) while repairing the remaining nick, an error-prone repair process (likely MMBIR normally triggered by DSB) has been recruited for unknown reasons, generating additional SVs or complex transposition events; and (3) the chance of MMBIR activation increases in non-B DNA.

Since duplications in flies are predominantly attributed to error-prone repair mechanisms rather than non-allelic homologous recombination (NAHR)[66], we examined 17 duplications in ISO1-1/2 (Fig. 3a), comprising 16 tandem duplications and 1 inverted duplication (Supplementary Fig. 10a). Consistently, only six (35.3%) show extensive (≥ 62 bp) breakpoint homology, suggesting NAHR. The remaining 11 duplications likely resulted from repair mechanisms (such as MMBIR) and exhibited error-prone features: (1) short de novo insertions (1–27 bp, median: 7 bp) at seven duplication terminals; (2) microhomology sequences at the other four duplications, including one triplication and one inverted duplication. Despite originating from distinct mechanisms, the sizes of duplication blocks are comparable (median: 4,518 bp vs. 3,651 bp, Supplementary Fig. 10a). Furthermore, duplications are enriched in intergenic regions (Supplementary Fig. 10b), a phenomenon likely influenced by both purifying selection and mutational bias. Specifically, TEs are overrepresented in intergenic regions and frequently form fragile clusters (Supplementary Fig. 10a)[69], serving as homologous anchors for NAHR and increasing the likelihood of DSBs and subsequent MMBIR.

Overall, the results indicate that error-prone DNA repair mechanisms contribute to transpositions, particularly complex transpositions in non-B DNA, as well as duplications.

## DNA- and RNA-level gene conversion of TEs contribute to 22.5% of clustered SNPs

In addition to SVs, one type of nucleotide variant, clustered single nucleotide polymorphisms (cSNPs defined as multinucleotide variants situated within 1–1000 bp of each other)[70], is also well-suited for long-read sequencing analysis but challenging for short-read analysis. Such mutations could result from numerous mechanisms, including gene conversion, secondary mutations, or error-prone DNA polymerase activity[71–73]. Given the pervasive TE conversion observed in human polymorphism data[74], we hypothesized that gene conversion of TEs significantly contributed to cSNPs in flies.

To test this hypothesis, we identified 1756 SNPs in the ISO1-1/2 assemblies (Fig. 4a, "Methods"). The patterns observed for SVs (Fig. 3a, b and Supplementary Data 5) are mirrored in SNPs: (1) the majority (1682 or 95.8%) of these SNPs were shared by both individuals and were more likely to be homozygous and covered by the two families (Figs. 4a, b and 2e); and (2) most (1420 or 84.4%) of these SNPs were again validated in aISO1-Anno (Supplementary Fig. 11). Hence, this SNP dataset also exhibits high quality. Among 1756 SNPs, 964 (54.9%) linked cSNPs were situated in proximity to each other (≤ 1000 bp; Fig. 4c, "Methods"). Following previous reports[70,71], we found a significant excess of cSNPs compared to the random expectation across four distance cutoffs (from 1–9 to 300–1000 bp, Fig. 4c). Furthermore, individual-specific cSNPs were located closer to each other than shared SNPs (Chi-square test $P = 0.010$), consistent with the disproportionate excess of cSNPs with a short distance in rare mutations[70]. In other words, as the population frequency increases (in the case of shared cSNPs), secondary mutations may occur, increasing the relative abundance of cSNPs at a long distance. The likelihood of this possibility is high since sequential mutations occur frequently[75].

We further analyzed how many cSNPs were generated by conversion. Specifically, we identified candidate conversion tracks associated with TEs by searching for cSNPs potentially contributed by paralogous donors (Supplementary Fig. 12, "Methods"). We identified 58 conversion events in TEs, encompassing 217 cSNPs (Fig. 4d). These numbers indicated that 32.5% (217/668) of the SNPs in TEs were generated by gene conversion. These 217 SNPs in TEs account for 22.5% of all 964 cSNPs and 12.4% of the total of 1756 SNPs we identified from ISO1-1 and ISO1-2. Note that two lines of evidence support the reliability of these conversion events. First, the median converted track length was 135 bp, aligning with the estimation based on a between-species analysis of flies[76]. Second, we observed the absence of a GC-biased conversion signal (Supplementary Fig. 13), in line with previous studies[77,78].

We investigated how these gene conversion events occur. Conversion generally occurs at the DNA level, where acceptors are converted by similar sequences in close chromosomal proximity[76,79]. However, in yeast, distance-independent conversion can occur at the RNA level, where TE transcripts serve as donors to convert homologous acceptors[80,81]. To test this possibility, we analyzed the chromosomal distance between acceptors and their corresponding donors ("Methods"). Among 34 conversions with unique donors, 15 (44.1%, Fig. 4e, the top panel) events occurred in close proximity (≤ 10 kb,

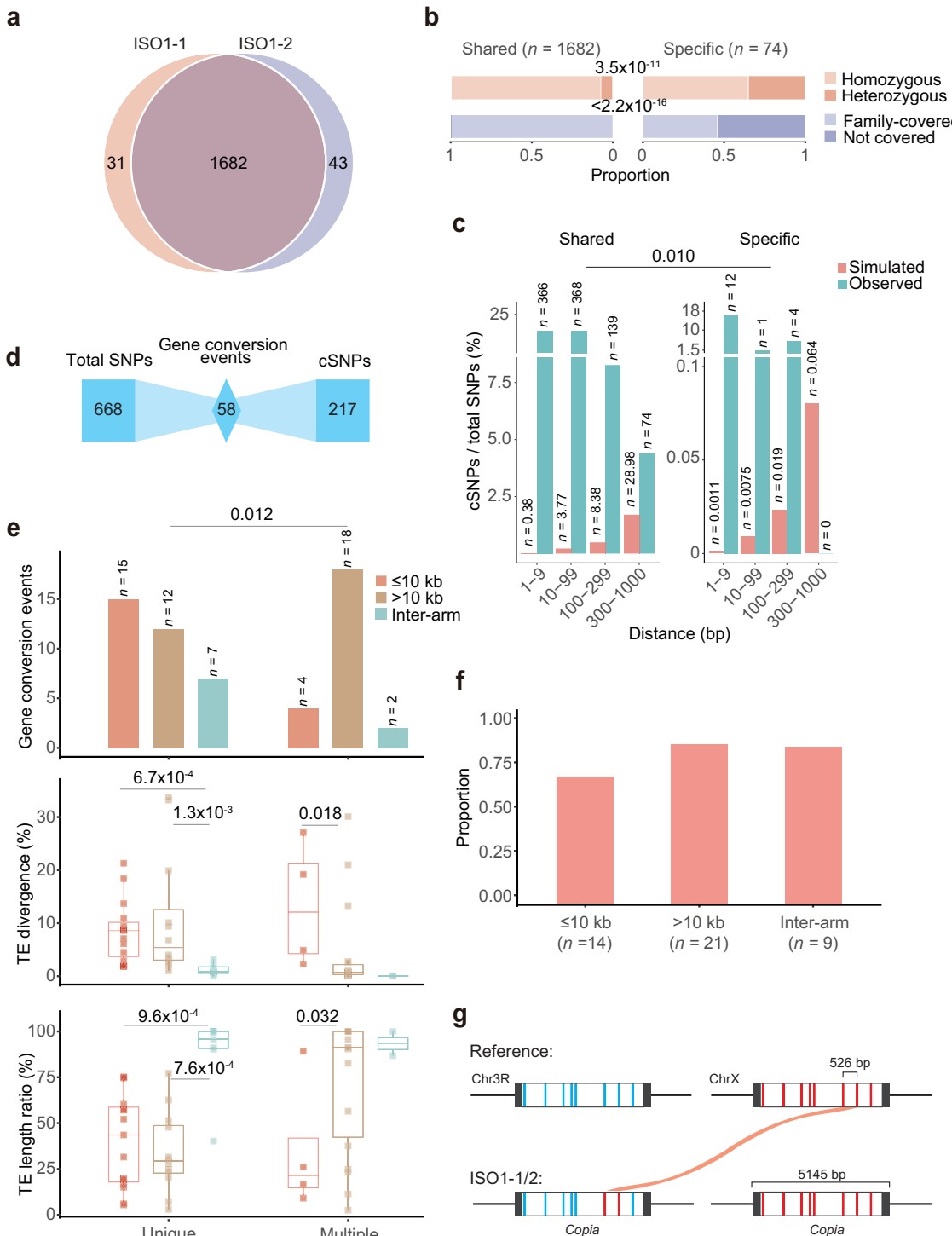

**Fig. 4 | TE conversion contributes to cSNPs. a** SNP count in ISO1-1/2. **b** Proportion of shared or individual-specific SNPs based on homozygosity and presence in fly families. **c** Inter-mutation distance distribution of cSNPs, with distance relative to the closest spaced SNPs. **d** TE conversion event count and corresponding cSNP count. **e** Characteristics of proximal, distant, and interarm conversion events. The top panel displays the counts of each gene conversion type, while the middle and bottom panels depict the sequence divergence and length coverage distributions of the donor relative to the consensus TE. The boxes show the first and third quartiles, with medians marked with the middle line, while whiskers show the minimum and maximum values. Sample sizes of the box plots are shown at the top

of the bar plots. Unique: gene conversion event with only one donor. Multiple: gene conversion event with multiple identical donors. For the middle and bottom panels, one-sided Wilcoxon rank-sum tests were performed for each pair, and only significant *P* values were shown. **f** A bar plot illustrating the proportion of TEs specifically expressed in spermatocytes. Note that this expression dataset covers the majority of TEs. **g** A potential RNA-level gene conversion example (#42 in Supplementary Data 6). Eight paralogous SNPs are marked in blue (acceptor) and red (donor). Long terminal repeats (LTRs) are marked in black, while the internal regions are shown in white. This conversion is shared by both ISO1-1 and ISO1-2.

"Methods"), 12 (35.3%) events were distant (>10 kb), and 7 (20.6%) events involved homologs harbored by different chromosomes or chromosomal arms. Since DNA-level gene conversion is believed to occur between similar sequences with distances of less than 10 kb[76,79], the latter two groups should be enriched for conversions occurring at the RNA level. Consistently, we identified a significantly higher proportion (20 or 83.3%, Fisher's exact test $P = 0.012$; Fig. 4e, the top panel) of distant and interarm gene conversion events in 24 events with multiple identical donors. This disparity should arise from the higher likelihood of remote conversions at both the DNA and RNA levels as the donor copy number increases.

RNA-level TE conversions necessitate the transcription of donor TEs[80], implying a young evolutionary age of these events. Consequently, distant and interarm donor TEs should exhibit low divergence, high structural integrity relative to consensus sequences, and increased germline expression. While the first two expectations generally held (Fig. 4e, the middle and bottom panels), the statistical significance was sometimes lacking, particularly for interarm gene conversion events with multiple identical donors, due to the small sample size ($n = 2$). A previous study showed that TEs tend to be expressed specifically in fly spermatocytes[82]. Donor TEs in distant and interarm gene conversion events display an even stronger tendency than proximal ones (Fig. 4f and Supplementary Data 6), despite non-significant test results.

A TE conversion event between two *Copia* copies provides a representative example of a potential RNA-level conversion. The acceptor resides on chr3R, while three candidate identical donors are encoded on chrX, chr3L, and chr2R. Taking the chrX-linked copy as an example, two of eight paralogous SNPs transfer from donor to acceptor, establishing a minimum conversion tract of 526 bp (Fig. 4g). All three candidate donors represent recent TE insertions, with divergences between 0.1–0.2% and TE consensus coverage of approximately 100% (Supplementary Data 6). Given the increased *Copia* expression observed in spermatocytes (Supplementary Data 6), an RNA-level conversion in the germline is plausible.

In summary, cSNPs are widespread, and a notable proportion of these SNPs are generated by DNA- and RNA-level TE conversion.

## Analyses of *Wolbachia* genomes demonstrate LILAP's effectiveness in dissecting symbiotic communities and elucidating mutational processes

Since up to 70% of insect species including *D. melanogaster* are infected by the endosymbiotic bacterium *Wolbachia pipientis*[83–86], we wondered whether we could assemble the *Wolbachia* genomes of the two individual flies and whether their mutations similarly reflect the properties of mutational processes observed in their host ISO1-1/2 genomes (Figs. 3, 4).

Before DNA extraction, we applied starvation treatment to eliminate the gut microbes of ISO1-1/2. As endosymbionts in fly cells, *Wolbachia* cannot be removed, leading to the coextraction and sequencing of fly and *Wolbachia* gDNA. To examine this situation, we taxonomically annotated hifiasm-assembled contigs by mapping them to sequences with taxonomic annotations ("Methods"). Most contigs were positively annotated to *D. melanogaster* (Fig. 5a). We identified one complete *Wolbachia* contig in each fly (hereafter named *w*ISO1-1 and *w*ISO1-2), corresponding to the complete *w*Mel reference genome of *Wolbachia* ("Methods"). In contrast, *w*Mel was assembled as five contigs in the previous single-fly sequencing data (Fig. 5b)[9].

Notably, unlike the results for ISO1-1, some contigs of ISO1-2 corresponded to bacterial genera such as *Acetobacter*, *Gluconopacter*, or *Candidimonas*, with a predominance of *Acetobacter* and *Gluconopacter* (Fig. 5a). Since *Acetobacter* and *Gluconopacter* are well-known fly midgut symbionts[87], hunger treatment seemed to effectively deplete gut microbes in ISO1-1 but not ISO1-2. Accordingly, the read depths for these bacteria were much lower than those for endosymbiotic *Wolbachia* (~5x vs. ~110x, Fig. 5a). Genera such as *Candidimonas* could also be symbionts, considering the similarity of their read depth and GC content to those of *Acetobacter* and *Gluconopacter*. Regarding the 94 contigs lacking taxonomic annotation, they likely correspond to unassembled Y-linked sequences of flies or the variable number of tandem repeats (VNTRs; Supplementary Data 7, "Methods"). Upon further scrutinizing the reference genome, we verified that 10 potential Y-linked sequences and three VNTRs could not be aligned back to the reference genome, indicating that they represent novel sequences. In total, these sequences amount to 181 kb, with an N50 of 27 kb (Supplementary Data 7).

Our analyses identified three mutations in *w*ISO1-1/2 relative to *w*Mel, including a 9-bp deletion and two point mutations, all of which were shared (Fig. 5b). The 9-bp deletion found in *GQX67_01850*, a one-unit contraction of a preexisting tandem repeat, likely stems from replication slippage (Fig. 5c)[88,89]. As *GQX67_01850* is a member of the ankyrin repeat gene family involved in bacteria-host interactions[83,84], this deletion could influence *Wolbachia*-fly dynamics. The point mutations occur in two *ltrA* genes (Fig. 5d): one generating a premature termination codon (PTC) in *ltrA_2* (the second *ltrA* copy alongside the *w*Mel genome) and another replacing Asp (D) as Asn (N) in *ltrA_7* (Fig. 5b). The nine *w*Mel *ltrA* copies form two phylogenetic groups with high between-group divergence (57.0%) and low within-group divergence (median: 0, Fig. 5d). *LtrA_2* and *ltrA_5*, in the same group, share an identical PTC mutation, suggesting gene conversion from *ltrA_5* to *ltrA_2*. Notably, *ltrA* encodes type II intron (a type of self-splicing retrotransposons) reverse transcriptase or maturase[90], suggesting that these two mutations may indicate functional constraints on TEs.

The analysis of these two point mutations highlights the abilities of long-read sequencing technology. As the total length of each *ltrA* locus reaches 2301 bp (*ltrA* 1548 bp; flanking paralogous region 753 bp; Fig. 5e), short-read sequencing cannot resolve *ltrA* paralogs within the same phylogenetic group (Fig. 5d). While the D->N change was previously identified in short-read data, it was unclear whether it represented a different allele or a paralogous difference[91]. Our long-read data revealed it to be a fixed SNP in *ltrA_7*, given the presence of Asn amino acid across all reads (Fig. 5e). Analogously, as with previous gene conversion analysis in ISO1-1/2 (Fig. 4), the identification of the PTC conversion from *ltrA_5* to *ltrA_2* was facilitated by long reads.

In summary, through the analyses of assembled LILAP data, we dissected symbiotic bacterial communities across two individual flies and identified mutational biases (repeat slippage and gene conversion of TEs) in *Wolbachia* genomes.

## Discussion

Herein, we developed LILAP, generated individual fly and endosymbiotic *Wolbachia* genome assemblies, and analyzed mutations that are challenging to study based on short-read sequencing. Consequently, this work holds significance across three dimensions, including method development, application potential, and the identification of mutational properties.

Specifically, LILAP builds upon previous low-input library preparation strategies but introduces critical modifications. Tn5-mediated tagmentation, which is widely used in low-input short-read sequencing library preparation and recently adopted in amplification-based long-read libraries[18,20,28,31,92], has been adapted for LILAP. LILAP has tailored tagmentation to enable DNA circularization using a hairpin adapter, further improving efficiency by digesting unbound adapters (Fig. 1c). In essence, LILAP efficiently generates circular DNA. This core innovation allows LILAP to construct low-input circular libraries for PacBio sequencers through a cost-effective and equipment-efficient process. Although we only tested LILAP with 100 ng of DNA, it holds the potential for even lower inputs in future experiments. At this stage, alternative methods, such as amplification-based approaches that can

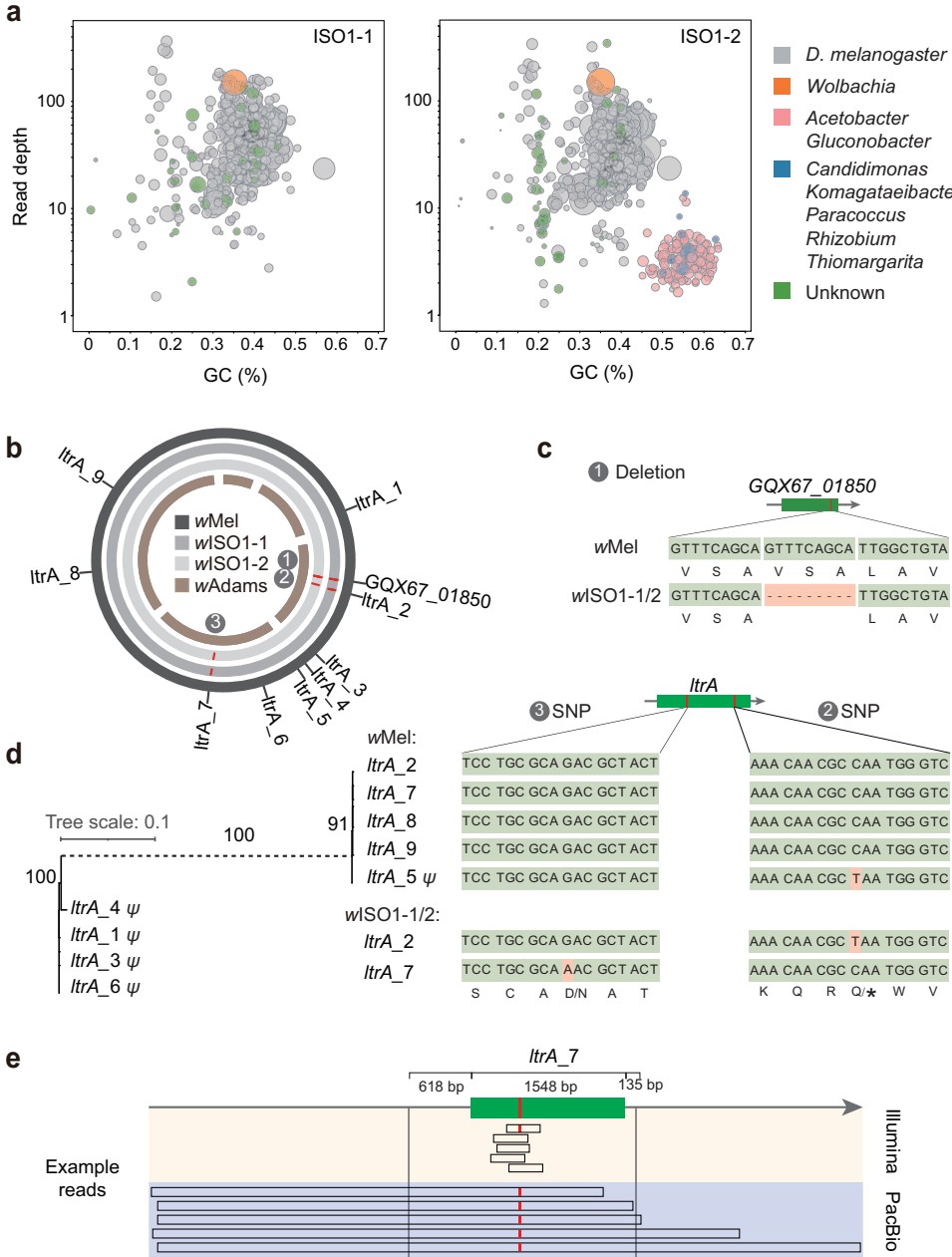

**Fig. 5 | Metagenomic information and mutations harbored by the *Wolbachia* genomes. a** Blobplot showing the read depth, GC content, and taxonomic origin of assembled contigs, with blob size proportional to the corresponding contig length. Contigs with an unknown taxonomic origin (green blobs) were identified as Y-linked contigs or VNTRs (Supplementary Data 7, "Methods"). **b** Circos plot illustrating the alignment of *w*Mel, *w*ISO1-1/2, and the previous assembly[9], marking three mutations and nine *ltrA*s in *w*ISO1-1/2. Five contigs were assembled in the ref. 9. **c** Deletion in *GQX67_01850*. Three amino acids (VSA) are removed in *w*ISO1-1/2. **d** Point mutations in *ltrA*_2 and *ltrA*_7. The left-side phylogenetic tree of the *ltrA* gene family, based on *w*Mel DNA sequences, shows bootstrap values around

internal nodes. The scale bar represents 10% nucleotide divergence. The substantial divergence of 57.0% between the two groups necessitates the representation of this branch as a dashed line. "ψ" denotes a pseudogene with at least one premature termination codon (PTC). **e** Schematic representation of read alignment at the *ltrA*_7 locus, with the D- > N mutation marked in red. The 1548-bp genic region together with 618-bp and 135-bp flanking regions of *ltrA*_7 are identical to those of the other four paralogous genes. A few Illustrative reads spanning the mutation are shown. Long reads align to this locus together with upstream or downstream regions, while short reads cannot differentiate this locus from paralogous loci.

accommodate 1–10 ng input (Fig. 1a and Supplementary Fig. 1b), could complement LILAP, expanding the scope of PacBio sequencing.

Furthermore, the high-quality individual fly genomes and complete *Wolbachia* genome indicate that LILAP could be adopted in four directions, particularly given the increasing data throughput and declining costs of PacBio sequencing[93]. First, LILAP will empower biodiversity exploration constrained by DNA input (Fig. 6a, the top panel), especially considering the close mass proximity of male flies to the median mass of insects (Fig. 1b). Although we have focused on fly

genomes (~180 Mb)[37,94], LILAP could conceivably enable the assembly of even larger genomes. In fact, with just 25% sampling, a decent fly assembly can still be achieved (e.g., gene completeness exceeding 98%; Supplementary Data 3). Therefore, it is feasible to assemble genomes as large as 500 Mb, especially those with low TE or repeat content, akin to *D. melanogaster*. Second, conventional genome-wide association studies (GWAS) of flies are based on short-read population sequencing[95], which can be confounded by incomplete genetic information and genetic diversity between individuals (Figs. 3a, 4a)[96,97].

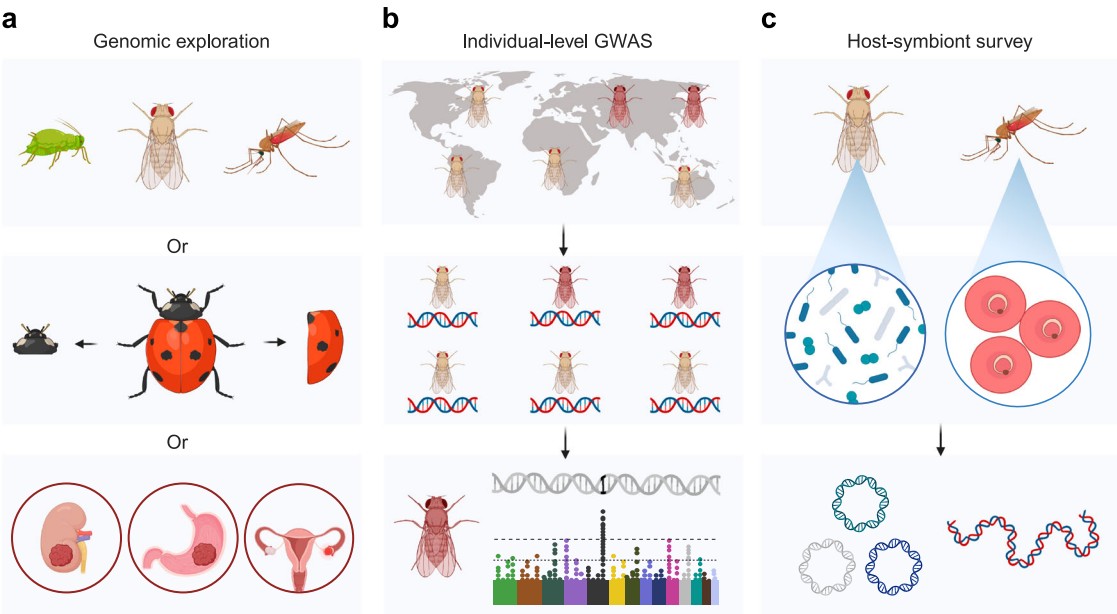

**Fig. 6 | Potential application directions of LILAP. a** LILAP for biodiversity exploration and somatic mutations. The top and middle panels depict the harvesting of DNA for sequencing from a whole small insect and a part of a large insect, respectively. The bottom panel illustrates the analysis of somatic mutations specific to certain tissues. **b** LILAP for individual-level GWAS analysis. The top, middle, and bottom panels demonstrate three stages of GWAS, including sampling, individual-level genotyping and phenotyping, and identification of association signals, respectively. **c** LILAP for the host-symbiont survey. The top and middle panels illustrate midgut symbionts and parasites, respectively, while the bottom panel shows circular and linear genomes of the microorganisms. Figure 6 was created with BioRender.com released under a Creative Commons Attribution-NonCommercial-NoDerivs 4.0 International license.

Individual genomes generated by LILAP effectively address this issue (Fig. 6b), although individual-level GWAS analysis was not conducted here due to the limited number of single flies and the extensive scope of topics covered in this study. Coupled with rapidly evolving high-throughput phenotyping techniques (e.g., automatic imaging analyses[98]), individual-level GWAS holds promise for efficiently unraveling the genotype-phenotype map. Third, since the *w*Mel strain is used to infect mosquitos and block viruses to control mosquito-borne diseases[91,99,100], the stability of *w*Mel genomes has been investigated by the short-read sequencing of multiple mosquitos[91,99]. LILAP will enable a more comprehensive dissection of *w*Mel and its host genomes, as well as other host-symbiont pairs, on the individual level (Fig. 6c). Fourth, many organisms, such as large insects or vertebrates, may yield more than 100 ng of DNA. For such cases, LILAP could exclusively utilize DNA extracted from specific body parts, such as wings or skin, to conserve materials (Fig. 6a, the middle and bottom panels). In this regard, LILAP could be also employed to investigate genomic differences within a single body, including somatic transpositions[101] or other types of SVs in both normal and tumor contexts[102].

An easily accessible computational framework for genome assembly and mutation annotation would enhance the widespread utilization of LILAP across all four aforementioned avenues. Consequently, we have made our codes publicly available, with the majority of them integrated into a single automated pipeline (see also Code availability). Except for the Tn5 binding sites filtering code, the remaining codes can also be applied to conventional PacBio sequencing data, which shares substantial similarities with LILAP data. Likewise, they can be adapted for amplification-based data, although additional codes are necessary to optimize the assembly of under-represented genomic regions (Fig. 2d) and to eliminate artificial SVs caused by template switching during amplification[22].

Finally, complex transpositions facilitated by non-B DNA and DNA- and RNA-level TE conversion underscores the capacity of LILAP to reveal mutational properties (Fig. 6). Similar to what has been found for *Jockey* elements (Fig. 3d, e and Supplementary Fig. 8), one non-LTR

retrotransposon in human, LINE1, is known to be associated with additional SVs[103]. In comparison, complex transpositions have not been reported for DNA TEs such as *hobo* elements. While non-B DNA has been linked to error-prone DNA repair[66,67,104], further investigations are needed to elucidate the pathway repairing transposition-associated nicks (Fig. 3g). On the other hand, although DNA-level TE conversions have been documented[72,79], RNA-level TE conversions have only been reported in yeasts[79–81]. Here, we provide evidence of their occurrence in *D. melanogaster* (Fig. 4e–g). Furthermore, short-read analyses indicate that conversion contributes to 10–20% of SNPs in human retrotransposons[74,105]. Our study provides a long-read-based estimate that TE conversion accounts for 32.5% of TE SNPs. Thus, TE conversion could be more widespread, especially considering the technical difficulty of short-read-based analyses[74,105]. Notably, since TEs can provide regulatory sequences, conversion provides an efficient means of disseminating linked beneficial mutations (as in the form of cSNPs) across TE copies[106,107].

In conclusion, we developed LILAP to reduce the required DNA input and cost of PacBio libraries. As a proof of principle, we assembled both individual fly and their endosymbiotic *Wolbachia* genomes, and characterized mutational properties. LILAP is therefore expected to foster the growth of fields including genomic analyses of small organisms or precious samples, and mechanistic studies targeting mutations recalcitrant to short-read sequencing.

## Methods

### Parameters of programs
All programs were run with the default parameters unless otherwise specified.

### Genomic DNA extraction
Single male adults (1–3 days old) of the *D. melanogaster* ISO1 strain were starved for 6 h[108] and then digested in 200 µl of lysis buffer [10 mM Tris-HCl, pH 8.0, 0.5% SDS, 100 mM EDTA and 1 mg/ml protease K (TIANGEN, RT403)] for >12 h at 37 °C. The DNA was then

extracted by Magnetic Animal Tissue Genomic DNA Kit (TIANGEN, DP341). Each male adult yielded ~100 ng of DNA.

## Transposome assembly

The single-strand adapter sequence consisted of Tn5 transposase binding sites and a PacBio adapter synthesized by Sangon:

5′-Phos-CTGTCTCTTATACACATCTATCTCTCTCTTTTCCTCCTC CTCCGTTGTTGTTGTTGAGAGAGAGATAGATGTGTATAAGAGACAG-3′ (Phos: phosphate modification). The first and last 19 bp were Tn5 binding sites, while the remaining sequence was the PacBio adapter. The sequence underwent annealing by heating at 95 °C for 5 min, followed by rapid cooling to room temperature in 1x TE buffer. A self-looping hairpin adapter at a concentration of 1 μM was formed. The final Tn5 transposome was constructed by combining 2 μl of Tn5 transposase (Mei5 Biotechnology, MF650-01), 2 μl of 10x TPS buffer, 6 μl of a 1 μM hairpin adapter solution, and 10 μl of ultrapure water, followed by a 30-minute incubation at 25 °C. The assembly of the transposomes for seven adapters with distinct barcodes (Supplementary Data 2) followed the same procedure.

## Library preparation for the single fly

The Tn5 transposome was combined with gDNA in T4 DNA ligase buffer (NEB, B0202S), followed by incubation at 55 °C for 10 min. The buffer contained 2 μl of 10x T4 DNA ligase buffer, 0.5 μl of Tn5 transposome mix, and 100 ng of single-fly DNA, with ultra-pure water added to reach a total volume of 20 μl. Subsequently, 0.25 μl of Exonuclease I (NEB, M0568L) and 0.25 μl of Lambda Exonuclease (NEB, M0262S) were added for the digestion of extra adapters at 37 °C for 15 min. The reaction was halted by adding 1 μl of 0.5% or 1% SDS buffer, followed by a 5-minute incubation. To counteract SDS effects, 1 μl of 20% Triton X-100 was added and allowed to act for an additional 5 min.

The tagged DNA underwent circularization by introducing 0.5 μl of T4 DNA polymerase (NEB, M0203L), 1 μl of T4 DNA ligase (NEB, M0202L) at 20 U/μl, 0.3 μl of 10 mM dNTPs, 1 μl of T4 DNA ligase buffer (NEB, B0202S), and 5 μl of ultra-pure water. This mixture was incubated at 12 °C for 20 min, followed by 30 min at 20 °C. For DNA repair, 1 μl of DNA repair mix comprising 0.1 μl of DNA Polymerase I (NEB, M0209S), 0.3 μl of UDG (NEB, M0280S), 0.15 μl of Fpg (NEB, M0240S), 0.15 μl of Endonuclease VIII (NEB, M0299S), 0.15 μl of Endonuclease IV (NEB, M0304S), and 0.15 μl of T4 PDG (NEB, M0308S) was added. The mixture was then incubated at 37 °C for 20 min. Finally, the library was purified using the Genomic DNA Clean & Concentrator-10 kit (ZYMO RESEARCH, D4010) for PacBio HiFi sequencing on the Sequel II platform. Approximately 8 GB of CCS reads were generated.

## Multiplexed sequencing library preparation

Two PacBio HiFi libraries were generated, each representing a distinct fly family composed of parents, two daughters, and two sons. Unique barcoded adapters were employed for indexing (Supplementary Data 1 and 2). To account for the difference in mass between female and male flies, a consistent DNA input of 100 ng was maintained for both sexes. Following adapter insertion and digestion, the resultant six sub-libraries were merged and purified via the Genomic DNA Clean & Concentrator-10 kit.

The multiplexed library then underwent circularization and DNA damage repair, following the process depicted in Fig. 1c. To eliminate noncircularized DNA, the DNA damage repair product was subjected to a 20-minute incubation at 75 °C, rendering the enzymes inactive. Subsequently, the circularized product was digested by introducing 0.25 μl of T7 Exonuclease (NEB, M0263L), 0.25 μl of Exonuclease III (NEB, M0206L), 0.25 μl of RecJf (NEB, M0264L), and 0.25 μl of Exonuclease I (NEB, M0568L) at 37 °C for 30 min. Notably, given the larger input of DNA, this step used more exonuclease compared to that described in the previous section to fully remove noncircularized DNA. The last step purified the final multiplexed library using the Genomic DNA Clean & Concentrator-10 kit.

## Compilation of single-fly sequencing data via amplification-based libraries

In alignment with ISO1-1/2 procedures, a single male adult fly was harvested for DNA extraction. With a DNA input of 40 ng, amplification-based sequencing libraries were constructed using the services of Annoroad™, employing the PacBio Ultra-Low DNA Input Kit (PN 101-987-800). For one PacBio Sequel II sequencing cell, 27 GB (190x) CCS data were generated. For comparative analysis, we downloaded 25 GB (179x) amplification-based CCS data from the PacBio company (NCBI SRR12473480), originating from a single female adult fly. This dataset was also generated with one cell of the Sequel II platform.

## Analysis of GC bias in the sequencing-depth distribution

CCS reads from single-fly LILAP libraries were processed as follows. After removing the 19-bp Tn5 binding sites, the CCS reads were mapped to the reference genome (UCSC dm6) with minimap2 v2.24[109]. To explore potential biases, controls included Tn5 tagmentation-based short-read data (NCBI SRA SRX7201057)[9], conventional PacBio bulk HiFi sequencing data (SRR10238607)[38], and the two aforementioned amplification-based sequencing datasets. The short-read data were mapped to the reference genomes (UCSC dm6) with BWA v0.7.17[110], while the PacBio data were mapped with minimap2. The GC content and sequencing depth were calculated within each 500-bp window[111]. The relative depth was defined as the ratio between the sequencing depth of each window and the median depth of all windows. For visualization, the plot illustrates the median relative depth corresponding to each GC content category, ranging from 20% to 65%. The visualization style in Fig. 2d follows the ref. 36.

## Genome assembly, polishing, and assessment

The raw long-read data in BAM format were processed with the CCS package v6.4.0 (https://github.com/PacificBiosciences/ccs) to generate CCS reads. All CCS reads with 19-bp terminal adapter sequences trimmed were subjected to de novo assembly by hifiasm v0.12[42]. The polishing pipeline, including Racon v1.6.0[112], merfin v1.1[113], and BCFtools v1.14[114], was used to automatically improve the primary assembly quality[41]. Next, we manually corrected assembly mistakes. First, the CCS reads were mapped to the primary assemblies using the repeat-aware aligner, Winnowmap2 v2.03[115] with the PacBio mode. The supplementary alignments, secondary alignments, and unmapped segments were excluded using SAMtools v1.17[116] command "view" with the parameter -F0x904. Second, given the high accuracy of Sniffles2[41,117], it (v2.0.6) was used to detect homozygous SVs supported by at least 3 CCS reads (--minsupport 3), i.e., assembly mistakes. We manually checked whether each homozygous SV involved reads mapped to two distinct chromosomal locations. We thus identified and rectified a misconnection between two contigs from chr3L and chrX in ISO1-1 and a misassembled insertion in ISO1-2. In addition, we conducted random downsampling of CCS reads to 15x coverage and employed the same assembly pipeline without polishing.

To be consistent with ISO1-1/2 sequencing data, CCS reads from amplification-based libraries were downsampled to a depth of 60x. Subsequently, we executed the assembly pipeline without conducting polishing procedures. For comparative purposes, we also performed assembly with all available 179 - 190x sequencing data.

Primary assemblies were used in the subsequent analysis unless otherwise specified since alternative assemblies outputted by hifiasm are small (9.3 Mb and 8.1 Mb) and mainly consist of repetitive sequences. The assemblies were evaluated by QUAST v5.2.0[118] and BUSCO v5.4.2[119] with the default parameters, except that the diptera_odb10 gene list was used for BUSCO[120]. We chose this single-copy

ortholog list because BUSCO analysis with orthologs shared by this specific lineage is more stringent.

Since Heavens et al.[10] did not release the assembled genome, we used Flye v2.9[121] to assemble the released ONT data. The assessment pipeline was the same as that of ISO1-1/2.

As in conventional analyses[55,122], dot plot analyses exclusively encompassed major chromosome arms, namely chr2L, chr2R, chr3L, chr3R, and chrX. All contigs of ISO1-1/2 following polishing and manual curation were aligned with the dm6 reference genome using minimap2 and then categorized by target arms. LG90 was accordingly calculated, while visualization of the genome-wide dot plot was achieved using D-Genies[123].

## QV calculation

Assembly-level QV was calculated with the reference-free quality assessment tool Merqury v1.3[124]. Given the 144-Mb fly reference genome, the k-mer size was set as 19 bp, as suggested by Merqury.

For CCS read generation, the minimum pass threshold in the CCS package was set as the default, i.e., 3. CCS reads with lengths between 3.5 kb and 6 kb were selected to calculate QV. If the CCS read had multiple mapping positions, we chose the hit with the longest mapping length including the highest number of matched bases. Note that we distinguished matches or mismatches based on the CIGAR tag of the BAM files generated by minimap2 with the parameter --eqx. As in the ref. [35], the concordance rate was calculated with the following equation: $Concordance = \frac{M}{M+X+D+I}$, where $M$ is the number of matches, $X$ is the number of mismatches, and $D$ and $I$ are the numbers of deleted and inserted bases in the CIGAR tag. QV is the Phred scale of Concordance: $QV = \min[-10 \times \log_{10}(1 - Concordance), -10 \times \log_{10}(1/(1 + Readlength))]$. In other words, QV is defined as the minimum of two values reflecting the possible lowest error rate associated with the given read length.

## *Sdic* gene family analysis

To ascertain the copy number and arrangement of the *Sdic* gene family, our initial step involved identifying the boundary genes *AnxB10* and *sw* within the genome assemblies using BLAST v2.12.0[125]. Thereafter, we extracted the contig of ISO1-1/2 that encompassed the complete *Sdic* gene family, positioned between *AnxB10* and *sw*. Employing BLAT v37x1[126], we searched for the reference *Sdic* gene copies against the contig. Given that these paralogs exhibited a divergence of at least 4%, as calculated with BLAST, we determined the organization of the six copies. In parallel, we extracted the sequences of each *Sdic* copy and the corresponding upstream intergenic region from the polished ISO1-1 primary assembly. Subsequently, pairwise BLAST analyses were conducted to determine their sequence identities, respectively.

We employed minimap2 (-ax map-hifi) to align reads against the primary assembly contigs and verified the sequencing coverage of the *Sdic* gene family locus. Supplementary alignments, secondary alignments, and unmapped segments were excluded using the SAMtools view command (-F0x904). The resultant alignment was subsequently sorted and indexed using the SAMtools sort and index commands, respectively. Snapshots of the ISO1-1/2 *Sdic* locus bam file in IGV v2.13.2[127] were presented in Supplementary Fig. 3b.

## SV analysis

By following the recent practice of calling SVs based on assemblies[128], we identified SVs in ISO1-1/2 and aISO1-Anno relative to the reference genome. In brief, both primary and alternative assemblies were aligned to the UCSC dm6 reference genome using MUMmer v4.0.0[129] and LASTZ v1.04.15[130]. These alignments were input into the svmu package (https://github.com/mahulchak/svmu) to detect SVs[55]. To verify SVs, custom Perl scripts were used to extract sequences containing SVs and up- and downstream 1 kb boundary sequences based on the svmu output. These sequences were aligned to the reference genome using

BLAT in the UCSC genome browser (https://genome.ucsc.edu/) to evaluate SV authenticity. To confirm TE insertions in ISO1-1/2 rather than TE excision in the reference genome, we manually examined candidate target site duplications (TSDs) for each TE insertion and confirmed the absence of TSDs in the latter genome. TE insertions in the reference genomes are similarly confirmed.

Since hifiasm produces a haploid assembly, heterozygosity information for SVs is absent. We leveraged the alignment file between dm6 and CCS reads, extracting reads mapped to SV loci. Sniffiles2 was used again to detect SVs (≥ 3 reads) based on CCS reads and obtained genotype and variant allele frequency (VAF) information for further heterozygosity analysis of SVs. For those SVs that could not be detected by Sniffiles2, VAF was calculated based on the BAM file. A binomial test was performed with an expected proportion of 100% since ISO1 represented an inbreeding line, and thus, the homozygosity was expected to be high. Cases that failed to reject the null hypothesis were deemed homozygous, while others were considered heterozygous. This test accounted for mapping errors and random depth fluctuations causing VAF deviations from 100% (homozygous) or 50% (heterozygous). However, six SVs failed to be genotyped because no read was long enough to span these loci. Given the small sample size, we simply excluded them in the subsequent analysis.

Individual-specific SVs were defined as those with reads supported by only one individual. If an SV was identified via assembly-level analysis in one fly but detected in another through read-level analysis, it was classified as a shared SV. Shared SVs, likely accumulated in the lab strain's ancestors, were also expected to be present in the two fly families, while the individual-specific SVs tended to be de novo and absent in the fly families. To test this hypothesis, we randomly chose 20 shared SVs spanning almost all types. The reasons that we selected 20 shared cases were that (1) downstream family analysis involves laborious manual curations, and (2) this number was comparable to that of individual-specific SVs ($n = 19$).

For multiplexed sequencing data, with Tn5 binding sites and barcodes removed, CCS reads were also aligned to the reference genome. To confirm the presence of the 39 identified SVs (20 shared and 19 individual-specific) in the two fly families, we employed IGV to visualize the read-mapping files. Through manual examination, we determined whether the respective loci contained these SVs.

We confirmed that these SVs were not artifacts induced by LILAP. Given the extensive manual curation efforts in SV identification, validation was carried out only on a subset of SVs utilizing amplification-based sequencing data from aISO1-Anno. A total of 89 SVs were chosen, including 80 randomly selected SVs (simple transpositions or other types of SVs) shared by ISO1-1 and ISO1-2, along with nine complex transpositions. Following the extraction of SV sequences from ISO1-1, they were mapped to the aISO1-Anno assembly, and CCS reads overlapping the focal sites were extracted. These reads were then aligned to the reference genome using BLAT, with subsequent manual curation conducted in the UCSC Genome Browser.

## TE insertion analysis across fly strains

Regarding the TE insertions in ISO1-1/2, we extracted the dm6-centric positions from the svmu output. These positions were compared with those of one ingroup strain (ISO1-BL) and three outgroup strains (CanS-SH, OreR-PB1, and w1118) retrieved from the TIDAL-fly database[25]. The result was visualized using an UpSet plot in TBtools-II v1.120137[131].

## Non-B DNA analysis

Up- and downstream of the flanking sequences surrounding the 180 simple TE transpositions and nine complex transpositions sites were extracted in both 200-bp and 1000-bp windows. Two control groups, each comprising 1000 fragments, one spanning 400 bp and another spanning 2000 bp, were randomly selected across the

genome. This step was executed using the "random" subcommand in BEDTools v2.31.0[132]. Non-B DNA motifs, including A_Phased_Repeat, G_Quadruplex_Motif, Direct_Repeat, Inverted_Repeat, Mirror_Repeat, Short_Tandem_Repeat, and Z_DNA_Motif, were predicted by non-B-gfa[133] with the default parameters, and the total count of all types of motifs was calculated in each sequence.

## SNP analysis

To perform SNP calling, we employed the recently developed assembly-level SNP identification method, specifically PAV v2.3.4[46]. This method facilitated the identification of SNPs within the ISO1-1/2 assemblies in comparison to the reference genome. To refine SNP filtering, we assessed PAV calling accuracy by manually inspecting SNPs on chr2L that were also supported by CCS reads. This revealed the possibility of misassembling repeat regions, even with long reads, prompting us to adopt a more stringent filtration approach. Specifically, we first defined repeat regions based on annotations from the RepeatMasker[134] and TRF (tandem repeat finder)[135] tracks on the UCSC genome browser. To filter PAV calls, we retained only those SNPs supported by sufficient uniquely mappable CCS reads. Supplementary, secondary, and unmapped reads were discarded. With the Python3 package Pysam v0.19.0[114], reads fully contained within repeat regions were removed, and those spanning at least one repeat boundary were retained. SNPs within VNTRs exceeding 1500 bp were also excluded due to potential assembly mistakes. Pileup files were generated using SAMtools. After excluding SNPs harbored by unanchored contigs in the current dm6 reference assembly, we further discarded SNPs that were mapped to more than one position, as indicated by a mapping quality of 0. In-house Python scripts were further developed to extract supporting reads and calculate VAF. Only SNPs supported by $\geq 3$ CCS reads, with a VAF $\geq 0.3$ on autosomes or VAF $\geq 0.9$ on sex chromosomes, were considered genuine. SNPs that were shared by ISO1-1/2 but were later identified as individual-specific due to filtering were excluded as potentially unreliable. Similar to SV analysis, SNP genotypes were determined using VAF and a binomial test.

SNPs of the two fly families were identified with SAMtools. Subsequently, whether the SNPs in ISO1-1/2 could be recovered in the families was analyzed.

The identical workflow, encompassing PAV and subsequent filters utilized for ISO1-1/2 assemblies, was applied to the aISO1-Anno assembly. An in-house Python script was utilized to automatically identify overlaps between SNPs in aISO1-Anno and ISO1-1/2, based on the consistency of their positions and mutation directions.

Clustered SNPs (cSNPs) were defined as those with at least two SNPs within a specific distance, both supported by the same reads. To identify cSNPs, we employed SAMtools "mpileup" command with the --output-QNAME parameter. In-house Python scripts were utilized to identify cSNPs within distance bins (1–9 bp, 10–99 bp, 100–299 bp, and 300–1000 bp), consistent with previous practice[70]. To assess the random occurrence of cSNPs, a Monte Carlo simulation was conducted using the "random" subcommand in BEDTools. We randomly sampled SNPs across major chromosomes and recorded the number of cSNPs within each bin, repeating this simulation 100,000 times to calculate the expectation.

## Detection of gene conversion

In principle, one conversion event could involve only one diagnostic SNP. However, to be conservative, we searched conversion events defined as those involving at least two linked SNPs[136]. To determine the boundaries of each conversion event, we initially clustered pairs of the nearest cSNPs with a distance of 1–1000 bp into sequence fragments (Supplementary Fig. 12). We then employed BLAT to search for potential conversion donors (perfect matches harboring the derived SNPs) for these fragments. For two cSNPs with a distance < 50 bp, we extended the search region symmetrically to 50 bp, ensuring a sufficiently long fragment for BLAT. The sequences were extracted using

SeqKit v2.4.0[137]. Following the first round of searching, we merged the fragments that had donors with the nearest SNP (clustered or single SNP within 1000 bp distance) as a query for a subsequent round of BLAT searching. This iterative process continued until it was impossible to merge fragments with the nearest SNPs, usually due to an inability to find a donor for the larger fragment. As in refs. 74,136,138, one acceptor could be matched to multiple identical donors, for which we chose the one with the minimum chromosomal distance to the acceptor. If all potential donors were encoded by other chromosomal arms, we randomly selected one.

The divergence values of TE copies were obtained from the UCSC RepeatMasker annotation file, and the TE coverage ratio was calculated using the following equation: $TE\ coverage = \frac{length\ of\ donor\ TE}{length\ of\ consensus\ TE}.$

Based on the distance between the donor and acceptor, we classified conversions as proximal ($\leq 10$ kb), distal ($> 10$ kb), or interarm (interchromosome or interarm) events. The rationale for choosing a cutoff of 10 kb was that gene conversion frequently occurs between homologs within $\leq 10$ kb[76,79].

The mutation direction of all SNPs at the acceptor site was examined to assess GC bias.

## *Wolbachia* genome and meta-genomic analyses

Using Hifiasm, we assembled the complete genome of *Wolbachia* as a single contig. The *D. melanogaster* reference strain ISO1 is known to harbor two different *Wolbachia* strains[139]: wMel (NCBI GCF_016584425) and wMelpop (NCBI GCF_016584325). We aligned the two assembled *Wolbachia* genomes (wISO1-1 and wISO1-2) to wMel and wMelpop using MUMmer and found that wMel was more closely related to the *Wolbachia* present in our flies. In comparison to wMel, we identified two SNPs and one indel in both *Wolbachia* genomes. We further remapped the CCS reads to the wMel genome, performed a manual check in IGV, and confirmed that all three mutations were fixed.

The *Wolbachia* genome was assembled as five contigs in a previous single-fly study[9]. Since the sequences are not publicly available, we estimated the lengths and coordinates of these five contigs given the original Fig. 3 in ref. 9. The Circos plot was generated using TBtools-II.

To perform taxonomic annotation and identify other potential prokaryotic content, we followed ref. 140 and used DIAMOND v2.0.13[141] together with NCBI BLAST v2.12.0 to search the contigs against UniProt proteins retrieved in April 2021[142] as well as the nucleotide collection (nt) database retrieved in February 2022[143]. With Blobtools2 v2.6.5[144], the results were processed, and bubble plots were generated.

Some of the 94 taxonomically unannotated contigs in ISO1-1/2 potentially belonged to unassembled Y-chromosome sequences. To identify Y-linked contigs, we separated multiplex family sequencing data into male and female groups and aligned these CCS reads to ISO1-1/2 contigs by using minimap2. The relative depth of each ISO1-1/2 contig was calculated as the median depth of the contig divided by the median depth of all contigs. Contigs meeting the criteria of having roughly half the relative depth (in comparison to known autosomal contigs) within the male group and a depth of zero in the female group were classified as Y-linked contigs.

Through manual curation, we identified the remaining 84 contigs as VNTRs. We utilized TRF v4.09[135] to generate consensus repeat units and observed that these VNTRs consist of hundreds to thousands of copies. This accounts for the failure of DIAMOND and BLAST to identify matches. Using BLAT, we successfully mapped 81 VNTRs back to the reference genome, while the remaining three VNTRs may not have been assembled in the reference.

## Figures and statistical tests

Depending on the context, we used bar, blob, box, Circos, line, scatter, UpSet, Venn, or violin plots to show the data distributions. In the boxplots, the hinges indicate first and third quartiles, and whiskers indicate minima and maxima. Values outside the 1.5 IQR (interquartile

range) are regarded as outliers. Simultaneously, the data points are overlaid on the figure. Violin plots are similar to boxplots except that the curves indicate the data density.

Depending on the specific plot type, the plots were generated with the R v4.2.1, ggplot2 v3.3.6, and ggtree v3.4.2.

Depending on the corresponding context, we tested significance with the binomial test, chi-square test, Fisher's exact test, or Wilcoxon rank-sum test. One-tailed tests were used.

## Reporting summary
Further information on research design is available in the Nature Portfolio Reporting Summary linked to this article.

## Data availability
The data supporting the findings of this study are available from the corresponding authors upon request. The LILAP and amplification-based aISO1-Anno sequencing data, along with the final assemblies generated in this study, have been concurrently deposited in the NCBI Bioproject database under accession code PRJNA983717 and in the National Genomics Data Center (part of the China National Center for Bioinformation) under accession code PRJCA019897.

Three public datasets used in this study are available in the NCBI Sequence Read Archive (SRA): amplification-based CCS data from the PacBio company under accession code SRR12473480, Tn5 tagmentation-based short-read data under accession code SRX7201057, and conventional PacBio bulk HiFi sequencing data under accession code SRR10238607. Source data are provided with this paper.

## Code availability
The codes produced in this study are available at https://github.com/Zhanglab-IOZ/LILAP and archived in Zenodo[145]. The codes for genome assembly, evaluation, polishing, and SV detection have been encapsulated as an integrated Snakemake v8.0.1[146] workflow. To facilitate user testing, demo data has also been provided. Users have the flexibility to execute the workflow as a whole or to run individual components as needed, given that both the Snakemake workflow and the raw codes are available. It is worth noting that codes for SNP calling or *Wolbachia* analyses were not integrated into the Snakemake workflow due to technical constraints (e.g., interaction with external data sources). Instead, the raw codes have been released on GitHub as well.

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

## Acknowledgements

We thank Drs. Xionglei He, Shujun Ou, Sihai Yang, Yang Shen, and Aihua Zheng together with Zhang lab members for their helpful discussions. This research was supported by the National Key R&D Program of China (2019YFA0802600 to Y.E.Z.), the National Natural Science Foundation of China (32325014 to Y.E.Z.), and the Chinese Academy of Sciences (ZDBS-LY-SM005 to Y.E.Z.).

## Author contributions

H.J., Y.E.Z., S.T., and J.R. conceived and designed the project. H.J. designed and developed LILAP, and prepared the sequencing libraries with the help of Y.Z. S.T. and Y.C. analyzed the data with the help of Y.G., J.S., H.M., Q.Z., J.C., and G.Q. Y.E.Z., H.J., S.T., Y.C., and J.R. wrote the manuscript. All authors read and approved the final manuscript.

## Competing interests

A patent (ZL202110042549.7) related to LILAP, filed by Y.E.Z., H.J., S.T., Y.C., and Q.Z. on January 13, 2021, was granted on May 17, 2022. The remaining authors declare no competing interests.
