## [Peer Review File · Nature Communications]

Low-input PacBio sequencing generates high-quality individual fly genomes and characterizes mutational processesREVIEWER COMMENTS

Reviewer #1 (Remarks to the Author):

Insects represent a sizable fraction of all known species. But the size of many of them, the inability to generate inbred lines, or their high levels of heterozygosity, make them refractory to their sequencing in terms of generating reference-like genome assemblies. This poses a serious challenge to global initiatives that aim to sequence every eukaryotic genome in our planet as a large fraction of eukaryotic species are tiny insects. Therefore, developing sequencing protocols that can be successful with low amounts of genomic DNA, ideally derived from a single individual, would represent a significant step forward in the field. This is the context in which Jia et al. have developed a new no amplification-based protocol for PacBio HiFi (LILAP), thus bypassing serious limitations in previous attempts (e.g. ref. 9). Their method is tested with two individual males of the reference strain ISO1 of *D. melanogaster*, which are infected with the endosymbiotic *Wolbachia*. The costs, speed, reproducibility, and no instrument dependency, make this new protocol particularly attractive. Further, the ability to reconstruct the *Sdic* multigene family provides a strong support for the quality of their approach, going beyond global metrics that cannot capture the challenge posed by structurally complex regions. In relation to the new knowledge obtained through this work, the authors report several key aspects about mutational events that are less likely to have been reliably detected and documented using short-reads-based assemblies. The observations include unique new TE insertions, some of which associated with local rearrangements, evidence of new transpositions occurring preferentially on non-B DNA regions in connection with error-prone DNA repair mechanisms, DNA- and RNA-level gene conversion among recent TE copies in connection with clusters of SNPs, and the capturing of symbiotic genomes associated with the flies sequenced.

Comments:

1. Table 1. As acknowledged by the authors, the N50 values of their assemblies are not particularly high compared to those from other assemblies, some of them obtained in fact with early chemistries. But beyond this detail, it would be useful for the reader to have another column in which the authors provide the number of contigs required to reconstruct the main 5 chromosomes (X, 2L, 2R, 3L, 3R) of *D. melanogaster*. If the main chromosomes are represented in 2-3 contigs maximum, I think it is safe to say that the outcome is particularly good.
2. The internal validity (the relative location and orientation of genes within scaffolds) has not been evaluated. I do not have any reason to think that it would be worse (in Fig. 3A, for example, there is no indication of inversions) than in other assemblies, but it would be nice to have this control done. The authors can generate whole-chromosome scale dot plots against the current reference genome assembly of ISO-1 and provide them as a supplementary figure.
3. Relative to the reconstruction of the *Sdic* region. The authors should acknowledge the annotation of the *Sdic* region presented in Clifton et al. 2017 (10.1093/molbev/msw212) using the assemblies in Berlin et al. 2015 and Kim et al. 2014. The last two papers never paid attention to the *Sdic* region and they did not reconstruct and annotate it. This should be clear in Table S3 and in the text, where the authors cite Clifton et al. 2020 instead of Clifton et al. 2017. In Clifton et al. 2020, the authors reconstruct and annotate the *Sdic* region in strains beyond ISO1. Further, it would be useful for the audience to indicate, at least in table S3, the number of reads that span the whole *Sdic* multigene family in full, including its flanking genes *AnxB10* and *sw*.
4. ISO1-BL. Are the authors referring to the strain used in Berlin et al. 2015? If so, please cite.
5. Observations about duplications are always provided in relation to transposable elements. Is there no DNA duplication unrelated to TEs? What is their size range? Are they all in intergenic regions?
6. This paper combines two very different components: a new low-input DNA sequencing protocol; and the characterization of mutational events, particularly some associated with TEs. The connection between both components is not properly explained. For example, why the authors did not attempt to track similar mutational events in other reference-like genome assemblies of *D. melanogaster* that have also used long-sequencing reads for scaffolding and were generated using genomic DNA from inbred lines, essentially reflecting 1 single individual? [For example 10.1038/s41467-019-12884-1] I

think this is the justification that should be provided to the readers and not the contrast with reference 25, as the author indicates in the last paragraph of the Introduction, or along the text in relation to short-read sequencing-based assemblies.

7. How the authors search for non-B DNA motifs should be described in greater detail.

8. Within discussion, when the authors talk about the scope of their sequencing approach in terms of generating genome assemblies, the number of reads and genome size are mentioned as key factors. Nevertheless, as a highly contiguous genome assembly is always one of the goals, the amount of repetitive DNA must be considered. In other words, even in the case of a genome size and sequence coverage as those shown by the authors for *D. melanogaster*, the level of contiguity could be lower than the one shown for their genomes in Table 1 if the number of repetitive sequences was substantially higher than that in *D. melanogaster*. Definitely, long sequencing reads can help bypass the "fragmentation" impact associated with repetitive sequences but it is not totally clear to me to what extent.

Reviewer #2 (Remarks to the Author):

The authors of the manuscript, titled "Low-Input long-read sequencing generates high-quality individual fly genomes and characterizes mutational processes", introduced LILAP, a novel library generation method specifically designed for PacBio technology. LILAP utilizes Tn5 transpose mediators and hairpin adapters to construct a circular PacBio sequencing library. In comparison to the amplification-based PacBio protocol, LILAP stands out due to its low-input requirement (100 ng), cost-effectiveness, and without amplification. In this study, the authors applied the LILAP library preparation method to sequence the genomes of two male *Drosophila melanogaster* individuals. Through a comprehensive analysis of the genomes and variants of these individuals, the authors demonstrated the efficacy of LILAP in investigating genome mutations, particularly in the context of small organisms and precious samples.

Here is a list of questions and concerns in no particular order and I hope these help improve the manuscript overall.

1. Since LILAP in this manuscript is specifically designed for PacBio technology, the title could be more specific, unless the authors can demonstrate the extension of LILAP to Oxford Nanopore technology.

2. Genome assembly performance is affected by sequencing depths. Thus, in Table 1, I suggest the authors generate similar depths of PacBio sequencing reads with the standard amplification-based PacBio protocol to perform a more meaningful comparison.

3. How many novel sequences do ISO1-1 and ISO1-2 have compared to the reference genome and other assemblies in this study? What is the N50 of these novel sequences?

4. In the SV analysis part, the SVs should be further validated by the standard amplification-based PacBio protocol to exclude the possibility that these mutations are introduced by the LILAP library preparation method.

5. What does the y-axis of the first panel in Fig 1b represent? Insects?

6. In Fig. 2c, could the centromere region be labelled?

Reviewer #3 (Remarks to the Author):

In this study, a low-input, low-cost, and amplification-free library preparation method is proposed for PacBio-based genome sequencing. The method is evaluated on two *Drosophila melanogaster* individuals and demonstrates multiple promising features, including high genome coverage, low sequencing errors (0.02%), and the potential to resolve structural variations and symbiont genomes. Overall, the method is impressive and has a wide range of applications. Before acceptance for publication, I have three minor questions.

1. The authors discuss the potential of individual-level GWAS with such a low-cost library preparation method, which is quite promising. My question is why the authors do not demonstrate a demo example of this interesting concept, given the advantages of this method. More details should be added or discussed on the promises and difficulties in implementing individual-level GWAS analysis.

2. The innovation of LILAP was stimulated by the need for insect studies. However, LILAP can be applied to more types of studies beyond insects. Therefore, the authors should show an example or add a discussion section to detail the potential applications (and potential limitations) outside of insect studies, given the ability of LILAP to derive both genome assembly and variations (0.02% error rate).

3. Genome assembly and annotation are currently still challenging. It is of great merit that the authors also share their code in Git Hub. A discussion section on how LILAP facilitates or imposes difficulties on genome assembly or annotation should be added. If an automatic workflow is available, it will shift the paradigm of genotype-phenotype studies.

Revision summary:

1. In response to Referee 1's comments, we have performed multiple analyses: demonstrated that a small number of contigs cover over 90% of chromosomal regions (LG90, Extended Data Fig. 5), supporting the decent contiguity of ISO1-1/2 assemblies derived from our LILAP (low-input, low-cost, and amplification-free library-generation method for PacBio sequencing); illustrated the absence of inversions with a genome-wide dot plot (Extended Data Fig. 6); presented intergenic duplication bias (Extended Data Fig. 10b); confirmed secondary mutations in previous non-ISO1 long-read sequencing data (Fig. R1 and Table R1), thereby highlighting the value of ISO1-1/2 in studying mutational mechanisms.
2. In response to Referee 2's comments, we have generated one (aISO1-Anno) and further downloaded another (aISO1-PB) amplification-based PacBio sequencing data to perform a head-to-head evaluation. ISO1-1/2 largely outperformed both amplification-based datasets (Fig. 2c-d and Table 1). Moreover, the majority of SVs and SNPs are validated in aISO1-Anno. We also identified 13 repeat-rich novel contigs.
3. In response to Referee 3's comments, we have expanded the Discussion section to elaborate on individual-level GWAS analysis, potential applications of LILAP beyond insects, and the comparison in computational frameworks between LILAP, conventional PacBio sequencing, and amplification-based PacBio sequencing (Fig. 6). In addition to sharing all raw codes, we have also developed an integrated workflow for sharing.

INDEX OF COMMENTS:

Comments of Reviewer #1 3

 Comment 1 3

 Comment 2 10

 Comment 3 11

 Comment 4 15

 Comment 5 16

 Comment 6 18

 Comment 7 21

 Comment 8 22

Comments of Reviewer #2 23

 Comment 1 23

 Comment 2 23

 Comment 3 29

 Comment 4 31

 Comment 5 33

 Comment 6 34

Comments of Reviewer #3 35

 Comment 1 35

 Comment 2 36

 Comment 3 37

Comments of Reviewer #1 (Remarks to the Author):

Insects represent a sizable fraction of all known species. But the size of many of them, the inability to generate inbred lines, or their high levels of heterozygosity, make them refractory to their sequencing in terms of generating reference-like genome assemblies. This poses a serious challenge to global initiatives that aim to sequence every eukaryotic genome in our planet as a large fraction of eukaryotic species are tiny insects. Therefore, developing sequencing protocols that can be successful with low amounts of genomic DNA, ideally derived from a single individual, would represent a significant step forward in the field. This is the context in which Jia et al. have developed a new no amplification-based protocol for PacBio HiFi (LILAP), thus bypassing serious limitations in previous attempts (e.g. ref. 9). Their method is tested with two individual males of the reference strain ISO1 of *D. melanogaster*, which are infected with the endosymbiotic *Wolbachia*. The costs, speed, reproducibility, and no instrument dependency, make this new protocol particularly attractive. Further, the ability to reconstruct the *Sdic* multigene family provides a strong support for the quality of their approach, going beyond global metrics that cannot capture the challenge posed by structurally complex regions.

In relation to the new knowledge obtained through this work, the authors report several key aspects about mutational events that are less likely to have been reliably detected and documented using short-reads-based assemblies. The observations include unique new TE insertions, some of which associated with local rearrangements, evidence of new transpositions occurring preferentially on non-B DNA regions in connection with error-prone DNA repair mechanisms, DNA- and RNA-level gene conversion among recent TE copies in connection with clusters of SNPs, and the capturing of symbiotic genomes associated with the flies sequenced.

Thank you for the thorough review of our manuscript and the detailed summary.

Comments:

Comment 1. Table 1. As acknowledged by the authors, the N50 values of their assemblies are not particularly high compared to those from other assemblies, some of them obtained in fact with early chemistries. But beyond this detail, it would be

useful for the reader to have another column in which the authors provide the number of contigs required to reconstruct the main 5 chromosomes (X, 2L, 2R, 3L, 3R) of *D. melanogaster*. If the main chromosomes are represented in 2-3 contigs maximum, I think it is safe to say that the outcome is particularly good.

Thank you for this constructive and insightful comment. It encompasses two distinct issues. First, we concur that contig numbers are informative, demonstrating that our ISO1-1/2 achieved a reasonable median LG90 (contig count covering over 90% of chromosomal arms) relative to comparable studies (4–5 vs. 1–86). Second, in our previous comparisons across different studies, we did not control for confounding technical factors, particularly the sequencing platform. Following the suggestion of Comment 2 of Referee 2, we included two sets of amplification data generated from a single fly on the same platform as ISO1-1/2, PacBio Sequel II HiFi. Once more, our ISO1-1/2 assemblies exhibited superior performance to the two assemblies based on amplification data across most evaluation parameters. As a result, we have updated Fig. 2c-d, Extended Data Figs. 5 and 6, as well as Table 1 accordingly.

The new version of the manuscript reads as:

Results:

“The remarkable coverage evenness likely stems from the amplification-free nature of LILAP or the absence of amplification bias²¹. To investigate this, we compiled two sets of individual fly sequencing data using standard amplification protocols on the PacBio Sequel II platform: one produced by Annoroad's commercial service (referred to as aISO1-Anno), and the other by PacBio itself (referred to as aISO1-PB, Materials and methods). As anticipated, both aISO1-Anno and aISO1-PB show more pronounced fluctuations (Fig. 2c), with low- or high-GC regions being underrepresented, displaying depths 10~80% lower than expected (Fig. 2d).” (Line 218-225)

“

Fig. 2 | Evaluation of LILAP sequencing data.

c, The chromosomal distribution of relative read depth with 100-kb window size. Annoroad's amplification-based sequencing data were denoted as aISO1-Anno, whereas PacBio's data were labeled as aISO1-PB. Except for the aISO1-PB, all other datasets were derived from male flies. Consequently, the presence of a minority of reads mapped to chrY in aISO1-PB likely indicates mismapping. Notably, the dots

positioned between chr2L (Left) and chr2R (Right), between chr3L and chr3R, and within chrX denote centromeres. **d**, The relative read depth in regions with different GC content. Standard PacBio HiFi sequencing data, Tn5 tagmentation-based short-read (Illumina) data, and amplification-based PacBio HiFi sequencing data served as controls. The left and right Y-axes show the relative depth and the proportion of the corresponding GC bins, respectively. The relative depth was calculated as $\frac{\text{depth of each window}}{\text{median (depth of all windows)}}$ (Line 183-200)

“Similarly, we conducted assemblies for two individual fly genomes using standard amplification protocols: aISO1-Anno and aISO1-PB. Subsequently, we assessed our final ISO1-1/2 assemblies against the newly generated aISO1-Anno/PB assemblies, alongside five previously published long-read fly assemblies. These include two individual fly assemblies^{9,10} and three assemblies generated by sequencing dozens or hundreds of flies^{38,43,44}. This ensemble of seven assemblies encompasses variations in sequencing platforms and depths, thereby providing a comprehensive control for evaluation.” (Line 244-251)

“The reference genome coverage exhibited even more pronounced disparities, with the ISO1-1/2 assemblies achieving 99.6% coverage for all major chromosomes, excluding the poorly assembled chrY⁴⁷⁻⁴⁹, whereas coverage for other assemblies varied between 80.9% and 99.2%.” (Line 262-266)

“Table 1. Evaluation of ISO1-1/2 assemblies and five previous long-read fly assemblies.

Study ^a	Platform ^b	Sequencing depth	DNA input ^c	Sex ^d	QV ^e	Genome coverage ^f	Gene coverage ^g	Size (Mb)	NG50 (Mb)	Median LG90 ^h
ISO1-1	PacBio	59x			60.6/64.6	99.6%	98.7%	164.1	6.7	5
ISO1-2	Sequel II HiFi mode	57x	100 ng	M	58.5/63.1	99.6%	98.7%	167.5	6.7	4
aISO1-Anno	PacBio	60x	40 ng	M	54.7	98.2%	97.9%	156.5	1.2	20
aISO1-PB	Sequel II HiFi mode	60x	10 ng	F	56.3	97.9%	98.7%	156.4	9.7	4
Berlin et al. ⁴³	PacBio Sequel II	90x	10 µg	M	42.9	99.2%	98.3%	164.1	13.6	2

CLR mode										
ONT+										
Adams et al. ⁹	Illumina+	60x	78 ng	F	33.6	80.9%	92.6%	111.0	21.7	NA
Hi-C										
Heavens et al. ¹⁰	ONT R9.4	36x	110 ng	Un	37.7	91.7%	94.4%	129.8	0.6	86
Solares et al. ⁴⁴	ONT R9.5	30x	1.5 µg	Mixed	28.5	82.8%	97.2%	140.7	2.9	11
PacBio										
Nurk et al. ³⁸	Sequel II	40x	>1 µg	F	63.3	97.2%	98.8%	147.3	20.2	1
HiFi mode										

- a. Adams *et al.* used a hybrid individual derived from the ISO1 and I38 strains⁹, while aISO1-PB and Nurk *et al.* used a hybrid population of ISO1 and A4³⁸. Other studies used the ISO1 strain exclusively. The assembled genome from Heavens *et al.* was not publicly available¹⁰, and we performed assembly independently (Materials and methods). In contrast, the other four studies made their assemblies public, and the statistics pertain to those assemblies.
- b. Adams *et al.* generated multiple sequencing data types from both the ONT and Illumina platforms⁹ and sequencing depth and DNA input values refer to the ONT data. Note that the ONT Flow Cell information is not available.
- c. The study by Nurk *et al.* lacks specific DNA quantity information³⁸. Nevertheless, considering the use of pooled flies, it is reasonable to infer that the DNA input exceeded 1 µg. DNA input for Adams *et al.* specifically refers to the ONT sequencing.
- d. “M”, “F” and “Un” refer to male, female, and unknown sexes, for which Heavens *et al.* did not provide specific information¹⁰.
- e. For ISO1-1/2 assemblies, QV values for both the initial and polished assemblies are presented, while for other assemblies, no further polishing was performed.
- f. To account for chromosomal differences between sexes, coverage was calculated across all chromosomes except chrY.
- g. Gene coverage was quantified using BUSCO complete gene models (Materials and methods).
- h. “Median LG90” was exclusively computed for chr2L, chr2R, chr3L, chr3R, and chrX. “NA” denotes the inability of the assembly from Adams *et al.* to cover 90% of the reference genome across the five arms.

The inclusion of two amplification-based assemblies (aISO1-Anno/PB) in our evaluation underscores the advantage of amplification-free practices. Despite utilizing the same sequencing platform (Sequel II HiFi) and similar sequencing depth (60x), both aISO1-Anno and aISO1-PB demonstrate comparatively lower genome coverage (98.2 and 97.9%, Table 1), potentially attributable to amplification bias (Fig. 2d). Consistent with this observation, they also exhibit relatively lower QV (55 and 56), likely stemming from amplification errors caused by PCR⁵⁰. Even with an increase in sequencing depth (from 60x to ~180x), these two metrics for both amplification-based assemblies exhibit only moderate changes (Supplementary Table 3), indicating that a sequencing depth of 60x suffices and that amplification bias or errors are not effectively mitigated by further depth augmentation.” (Line 275-312)

“This results from the lack of an independent scaffolding method (*e.g.*, Hi-C⁹) and a short insert size (4.5 kb vs. 8.2 kb or higher, Fig.2a and Extended Data Fig. 4). However, even for this parameter, ISO1-1/2 showed a value of 6.7 Mb (Table 1), which is better than aISO1-Anno (1.2 Mb) and the two ONT-based assemblies (0.6–2.9 Mb). Consistently, their median LG90s (number of contigs covering over 90% chromosomes) are only 4 or 5 (Table 1 and Extended Data Fig. 5), showcasing a reasonable level of continuity.

Collectively, the results indicated that the ISO1-1/2 assemblies largely outperformed other long-read assemblies from single or pooled flies, achieving a base error of 10⁻⁶ and near-complete coverage of reference genomes or genes.” (Line 329-338)

Materials and methods:

“Compilation of single-fly sequencing data via amplification-based libraries

In alignment with ISO1-1/2 procedures, a single male adult fly was harvested for DNA extraction. With a DNA input of 40 ng, amplification-based sequencing libraries were constructed using the services of Annoroad™, employing the PacBio Ultra-Low DNA Input Kit (PN 101-987-800). For one PacBio Sequel II sequencing cell, 27 GB (190x) CCS data were generated. For comparative analysis, we downloaded 25 GB (179x) amplification-based CCS data from the PacBio company (NCBI SRR12473480),

originating from a single female adult fly. This dataset was also generated with one cell of the Sequel II platform.” (Line 804-812)

“To be consistent with ISO1-1/2 sequencing data, CCS reads from amplification-based libraries were downsampled to a depth of 60x. Subsequently, we executed the assembly pipeline without conducting polishing procedures. For comparative purposes, we also performed assembly with all available 179~190x sequencing data.” (Line 845-848)

“As in conventional analyses^{55,122}, dot plot analyses exclusively encompassed major chromosome arms, namely chr2L, chr2R, chr3L, chr3R, and chrX. All contigs of ISO1-1/2 following polishing and manual curation were aligned with the dm6 reference genome using minimap2 and then categorized by target arms. LG90 was accordingly calculated, while visualization of the genome-wide dot plot was achieved using D-Genies¹²³.” (Line 859-864)

“Data availability

The LILAP and amplification-based aISO1-Anno sequencing data, along with the final assemblies produced in this study, have been concurrently submitted to the NCBI Bioproject database under accession number PRJNA983717 and to the National Genomics Data Center (part of the China National Center for Bioinformation) under accession number PRJCA019897.” (Line 1077-1082)

Extended Data Figure:

Extended Data Fig. 4 | The length distribution of CCS reads for amplification-based PacBio sequencing data.

The figure convention follows Fig. 2a. ”

Extended Data Fig. 5 | Number of contigs covering over 90% (LG90) of five main arms.

Distribution of LG90 across main arms for ISO1-1 (a) and ISO1-2 (b). Grey bars with gradients denote contigs ordered by sizes, while white bars depict unassembled sequences in the reference genome.”

Comment 2. The internal validity (the relative location and orientation of genes within scaffolds) has not been evaluated. I do not have any reason to think that it would be worse (in Fig. 3A, for example, there is no indication of inversions) than in other assemblies, but it would be nice to have this control done. The authors can generate whole-chromosome scale dot plots against the current reference genome assembly of ISO-1 and provide them as a supplementary figure.

Thanks. We generated dot plots and identified genome-wide collinearity without any inversions.

The new version of the manuscript reads as:

Results:

“Despite the global collinearity between ISO1-1/2 and the reference genome (Extended data Fig. 6), we identified 337 SVs in ISO1-1/2 (Fig. 3a). All these SVs involved structural alterations of known sequences present in the reference genome, such as TE insertions or duplications.” (Line 343-346)

Materials and methods:

“As in conventional analyses^{55,122}, dot plot analyses exclusively encompassed major chromosome arms, namely chr2L, chr2R, chr3L, chr3R, and chrX. All contigs of ISO1-1/2 following polishing and manual curation were aligned with the dm6 reference genome using minimap2 and then categorized by target arms. LG90 was accordingly calculated, while visualization of the genome-wide dot plot was achieved using D-Genies¹²³.” (Line 859-864)

Extended Data Figure:

“

Extended Data Fig. 6 | Dot plots between ISO1-1/2 contigs and the five main chromosome arms of the reference genome.

Black lines in the left margin and corresponding dashed lines denote contig boundaries, while colors indicate four identity ranges. It is noteworthy that regions with low identity (*e.g.*, green) mapped by numerous short contigs predominantly correspond to repeat-rich centromeres (refer also to Fig. 2c).”

Comment 3. Relative to the reconstruction of the Sdic region. The authors should acknowledge the annotation of the Sdic region presented in Clifton et al. 2017 (10.1093/molbev/msw212) using the assemblies in Berlin et al. 2015 and Kim et al. 2014. The last two papers never paid attention to the Sdic region and they did not reconstruct and annotate it. This should be clear in Table S3 and in the text, where

the authors cite Clifton et al. 2020 instead of Clifton et al. 2017. In Clifton et al. 2020, the authors reconstruct and annotate the *Sdic* region in strains beyond ISO1. Further, it would be useful for the audience to indicate, at least in table S3, the number of reads that span the whole *Sdic* multigene family in full, including its flanking genes *AnxB10* and *sw*.

Thanks for this insightful comment. We have rectified the citations in the text, Table S3, and the legend of Table S3. Moreover, since the majority of CCS reads are approximately 4.5 kb (Fig. 2a), they cannot span the entire *Sdic* locus (~57 kb). The reason that we were able to fully resolve the *Sdic* locus is due to our precise (>99.9% accuracy) CCS reads containing adequate sites that differentiate copies with identities lower than 99%. Consequently, we have updated both the Results and Materials and methods sections accordingly.

The new version of manuscript reads as:

Results:

“This locus is difficult to assemble due to high sequence similarity between paralogous genes or paralogous intergenic regions, with similarities reaching up to 96% and 99%, respectively (Extended Data Fig. 3a). These complexities, combined with sequencing errors, have led to conflicting reports, showing four, seven, or six copies, among which six copies have been proven to be correct⁵². With our CCS reads averaging ~4.5 kb in length (Fig. 2a) and exhibiting a low sequencing error rate (0.02%, Fig. 2b), each read is anticipated to contain an ample number of authentic sites capable of distinguishing between paralogous regions. Coupled with adequate sequencing depth (Extended Data Fig. 3b), we achieved the successful assembly of a single contig in ISO1-1/2 assemblies, with all six copies arranged in the correct order (Supplementary Table 4).” (Line 315-326)

Materials and methods:

“Given that these paralogs exhibited a divergence of at least 4%, as calculated with BLAST, we determined the organization of the six copies. In parallel, we extracted the sequences of each *Sdic* copy and the corresponding upstream intergenic region from the polished ISO1-1 primary assembly. Subsequently, pairwise BLAST analyses were conducted to determine their sequence identities, respectively.

We employed minimap2 (-ax asm5) to align reads against the primary assembly contigs and verified the sequencing coverage of the *Sdic* gene family locus. Supplementary alignments, secondary alignments, and unmapped segments were excluded using the SAMtools view command (-F0x904). The resultant alignment was subsequently sorted and indexed using the SAMtools sort and index commands, respectively. Snapshots of the ISO1-1/2 *Sdic* locus bam file in IGV v2.13.2126 were presented in Extended Data Fig. 3b.” (Line 887-898)

Extended Data Figure and Supplementary Table:

“

Extended Data Fig. 3 | Analysis of the *Sdic* locus.

a, Nucleotide sequence identity of various *Sdic* genes or the upstream intergenic loci.

b, Distribution of sequencing depth and read alignment. It is noteworthy that a single contig (~6.7 Mb) in both ISO1-1 and -2 assemblies contains *Sdic* copies in the correct number and order.”

“

Assembly	Copy number	Order (telomere...<- <- AnxB10 ...<- <- sw ...centromere)
UCSC dm5	4	AnxB10 -1-2-3-4- sw
UCSC dm6	7	AnxB10 -1-2-3-A-B-C-4- sw
Berlin et al. , 2015 (resolved by Clifton et al. , 2017)	6	AnxB10 -1-4-3-B-C-2- sw
Kim et al. , 2014 (resolved by Clifton et al. , 2017)	6	AnxB10 -1-4-3-B-C-2- sw
ISO1-1	6	AnxB10 -1-4-3-B-C-2- sw
ISO1-2	6	AnxB10 -1-4-3-B-C-2- sw

Supplementary Table 4. Copy number and gene arrangement of *Sdic* locus among different assemblies of the ISO1 strain.

As a tandem gene family, *Sdic* is positioned on chrX, flanked by *AnxB10* and *sw*. In the ISO1 reference genomes, releases five (UCSC dm5) and six (UCSC dm6), this gene family is represented by four and seven copies, respectively. Conversely, all other four genome assemblies contain six copies in an identical order. In addition, in dm5 and dm6, the positions of *Sdic2* and *Sdic4* are exchanged compared to other assemblies. Notably, Berlin *et al.* (2015) and Kim *et al.* (2014) sequenced the same ISO1 subline used for the *Drosophila* Genome Project, which produced the reference genome. Despite generating the original sequencing data, they did not resolve the *Sdic* locus. However, Clifton *et al.* (2017) successfully resolved it in both projects.”

Comment 4. ISO1-BL. Are the authors referring to the strain used in Berlin et al. 2015? If so, please cite.

Thanks. According to the literatures^{1,2}, ISO1-BL represents a recent isolate of the ISO1 strain obtained from the Bloomington *Drosophila* Stock Center, whereas the ISO1 strain in Berlin *et al.* (2015) is a subline of the reference strain utilized by the *Drosophila* Genome Project. However, it is worth noting that they share a "close relationship"². We have clarified this issue in the legend of Supplementary Table 4 (please refer to our response to Comment 3) and the legend of Extended Data Fig. 7a.

Extended Data Fig. 7 | SVs in ISO1-1/2 and DSPR strains.

a, An UpSet plot showing the distribution of TE insertions across different lab fly strains. For example, 1,244 insertions (shown as a truncated bar) are specific to OreR-PB1, while 65 insertions are only present in OreR-PB1 and w1118. **Note that ISO1-BL represents a recent isolate of the ISO1 strain obtained from the Bloomington *Drosophila* Stock Center.**

Comment 5. Observations about duplications are always provided in relation to transposable elements. Is there no DNA duplication unrelated to TEs? What is their size range? Are they all in intergenic regions?

Thanks. To ensure the manuscript's conciseness, we previously focused on TE insertions. However, as shown in Fig. 3a, we indeed detected 17 duplications. Motivated by your comment, we analyzed these duplications from the aspect of mutational mechanisms, which were also the major scope of the manuscript. In brief, both repair mechanisms and homologous recombination seem to be involved in the generation of these duplications. Despite distinct mechanisms, they show similar median size (~4 kb). Possibly driven by mutational bias and selection force, they are indeed enriched in intergenic regions.

The new version of the manuscript reads as:

Results:

“Since duplications in flies are predominantly attributed to error-prone repair mechanisms rather than non-allelic homologous recombination (NAHR)⁶⁶, we examined 17 duplications in ISO1-1/2 (Fig. 3a), comprising 16 tandem duplications and 1 inverted duplication (Extended Data Fig. 10a). Consistently, only six (35.3%)

show extensive (≥ 62 bp) breakpoint homology, suggesting NAHR. The remaining 11 duplications likely resulted from repair mechanisms (such as MMBIR) and exhibited error-prone features: 1) short *de novo* insertions (1–27 bp, median: 7 bp) at seven duplication terminals; 2) microhomology sequences at the other four duplications, including one triplication and one inverted duplication. Despite originating from distinct mechanisms, the sizes of duplication blocks are comparable (median: 4,518 bp vs. 3,651 bp, Extended Data Fig. 10a). Furthermore, duplications are enriched in intergenic regions (Extended Data Fig. 10b), a phenomenon likely influenced by both purifying selection and mutational bias. Specifically, TEs are overrepresented in intergenic regions and frequently form fragile clusters (Extended Data Fig. 10a)⁶⁹, serving as homologous anchors for NAHR and increasing the likelihood of DSBs and subsequent MMBIR.” (Line 447-462)

Extended Data Figure:

a

Region	Size (bp)	Genomic region	Homology (bp)	Mechanism
chr2R:4272799-4284616	11818	Intergenic	3416	NAHR, TE-rich region
chr3L:15296254-15298609	2356	Intergenic	437	NAHR, TE-rich region
chr3L:25382610-25389303	6679	Intron	376	NAHR, TE-rich region
chr3R:2707715-2720679	12967	Intergenic	12961	NAHR, TE-rich region
chr3R:1529607-1530269	663	Intron	663	NAHR, TE-rich region
chrX:14512658-14513007	359	Intergenic	62	NAHR, TE-rich region
chr2L:8976298-8978276	59	Intergenic	-27	Error-prone DNA repair
chr2L:16137177-16141792	4629	Intergenic	-14	Error-prone DNA repair
chr2L:17988362-17992012	3651	CDS	1	Error-prone DNA repair, triplication
chr2R:5394675-5401717	7056	Intergenic	-13	Error-prone DNA repair
chr2R:16081776-16084912	3137	CDS	-1	Error-prone DNA repair
chr3L:24193977-24200069	6093	Intron	1	Error-prone DNA repair
chr3L:24871933-24881238	9266	Intergenic	-2	Error-prone DNA repair, TE-rich region
chr3L:26051334-26055944	4611	Intergenic	3	Error-prone DNA repair, TE-rich region
chr3R:16794301-16794715	418	Intergenic	-7	Error-prone DNA repair
chr3R:23516152-23516304	174	Intron	21	Error-prone DNA repair, inverted duplication, TE-rich region
chrX:186648-187262	618	Intron	-5	Error-prone DNA repair

b

“

Extended Data Fig. 10 | Duplications in ISO1-1/2.

a, Summary of the 17 duplications mediated by different mechanisms. Negative values in the "Homology (bp)" column indicate *de novo* insertions at the middle breaking point of two duplication blocks. Homology sequences ranging from 1 to 30 bp were considered as microhomology⁵. "CDS" denotes coding sequence. **b**, Relative proportion of sequences across genomic regions."

Comment 6. This paper combines two very different components: a new low-input DNA sequencing protocol; and the characterization of mutational events, particularly some associated with TEs. The connection between both components is not properly explained. For example, why the authors did not attempt to track similar mutational events in other reference-like genome assemblies of *D. melanogaster* that have also used long-sequencing reads for scaffolding and were generated using genomic DNA from inbred lines, essentially reflecting 1 single individual? [For example 10.1038/s41467-019-12884-1] I think this is the justification that should be provided to the readers and not the contrast with reference 25, as the author indicates in the last paragraph of the Introduction, or along the text in relation to short-read sequencing-based assemblies.

Many thanks for this insightful comment. We chose not to analyze previous long-read assemblies, such as 10.1038/s41467-019-12884-1³, because these non-ISO1 *Drosophila* synthetic population resources (DSPR) lines exhibit a considerable evolutionary distance from the reference genome of ISO1. Consequently, analyzing mutation mechanisms in these lines could be confounded by secondary mutations. Consistently, the ref.³ explicitly stated "12.8% SV mutations, for which mutation annotation were complicated by secondary mutations". Among these 12.8% instances, we examined 10 cases, uncovering six unequivocal secondary mutation events, while the remaining four cases represent mutational hotspots featuring recurrent mutations. Given that all these scenarios involve multiple SVs at identical loci, the likelihood of secondary mutations should increase further when SNPs or indels are also taken into account.

We presented our curation results here while modifying the main text to enhance the logical flow.

A manual inspection of 10 cases revealed six instances where a TE insertion was followed by insertion or deletion within the TE. Two examples are shown below.

Fig. R1 | Alignments of contigs from the 13 DSPR strains against the reference genome, showing TE insertions followed by either another nested TE insertion (**a**) or a deletion (**b**). Blue and red bars represent forward and complementarily aligned contigs, and orange bars represent insertions with numbers indicating the lengths (bp). Using Panel **b** as an example, a secondary recombination event between the long terminal repeats (LTRs) of one *Gypsy* element results in a solo LTR, likely originating in the common ancestor of A6 and B4 strains. Conversely, B1 represents the ancestral state with the intact *Gypsy* element.

All 10 cases are shown here.

REF_chr	Position	Strains with the first mutation	Types of the first mutation	Strains with the secondary mutation	Type of the secondary mutation
chr2L	5771473	A2, A7, B3	TE insertion	A2	TE insertion into TE
chr2L	5724341	A4, A5, A7, B3, AB8	TE insertion	A5	Deletion within TE
chr2L	13139044	A7, B3	TE insertion	B3	Deletion within TE
chr2R	10014840	B3, AB8	TE insertion	B3	Deletion within TE
chr2R	18514168	A4, A5	TE insertion	A5	Deletion within TE
chr3R	17095916	A6, B1, B4	TE insertion	A6, B4	Deletion within TE
chr2L	17365202	A6, B3	Different types of TE insertion		Recurrent mutation
chr3L	11390568	B3, B4	Different types of TE insertion		Recurrent mutation
chr3R	25464660	B3, B4	The same type of TE with different non-nested sequence content		Recurrent mutation
chrX	2781273	A2, A6	The same type of TE with different non-nested sequence content		Recurrent mutation

Table R1. Manual curation of 10 potential secondary mutations in the 13 DSPR strains.

The table presents 10 potential secondary mutations identified in the study³. "Recurrent mutation" denotes four cases where multiple mutations occurred with unclear mutational order. For two cases, the same genomic positions harbor different types of TE insertion. For the remaining two cases, the same genomic positions harbor the same type of TE insertion but with different non-nested sequence content, hence also classified as "Recurrent mutation".

Materials and methods are shown here.

In the ref.³, a total of 365 SVs were annotated as complicated with secondary mutations ("complex" or CE = 2 in the mutation annotation file). To confirm that they indeed represent secondary mutations, we sampled 10 SVs for manual examination. That is, we generated the BAM files by aligning their contigs to the dm6 reference genome with BLAT and examined their alignment boundaries in the UCSC Genome Browser.

The new version of the manuscript reads as:

Introduction:

“We evaluated LILAP in two individuals of the ISO1 reference strain based on three considerations: 1) the two aforementioned works tested their methods in flies^{9,10}, making method comparison feasible; 2) male flies, given their proximity to the median body mass for insects^{23,24}, representing this lineage (Fig. 1b); and 3) mutations carried by individual ISO1 flies tend to be recently generated, providing a more accurate reflection of mutational properties compared to relatively older mutations found in a pool of individuals or in non-ISO1 flies, which are likely

confounded by secondary mutations^{25,26}. We therefore *de novo* assembled two individual genomes, resulting in substantially improved quality compared to previous individual-based assemblies. To further demonstrate the value of LILAP, we analyzed these two high-quality genomes and identified two novel mutational properties in which non-B DNA structures are associated with complex transposition events [transposable element (TE) insertions with extra duplications and/or deletions] and DNA- or RNA-level gene conversion events homogenize TEs.” (Line 78-91)

Results:

“The latter pattern gets more pronounced when considering all types of SVs in the published genomes of 13 *Drosophila* synthetic population resources (DSPR) founder strains (median: 3,241, Extended Data Fig. 7b)²⁶. The substantial evolutionary divergence between DSPR strains and ISO1 accounts for 12.8% SVs, which were complicated by secondary mutations²⁶.” (Line 362-366)

“

Extended Data Fig. 7 | SVs in ISO1-1/2 and DSPR strains.

b, Counts of euchromatic SVs across 13 DSPR founder strains. The data were obtained from https://github.com/mahulchak/dspr-asm/tree/master/variant_data.”

Comment 7. How the authors search for non-B DNA motifs should be described in greater detail.

Thanks. The Materials and methods section has been elaborated to encompass

additional details.

“Non-B DNA analysis

Up- and downstream of the flanking sequences surrounding the 180 simple TE transpositions and nine complex transpositions sites were extracted in both 200-bp and 1,000-bp windows. Two control groups, each comprising 1,000 fragments, one spanning 400 bp and another spanning 2,000 bp, were randomly selected across the genome. This step was executed using the "random" subcommand in BEDTools v2.31.0¹³¹. Non-B DNA motifs, including A_Phased_Repeat, G_Quadruplex_Motif, Direct_Repeat, Inverted_Repeat, Mirror_Repeat, Short_Tandem_Repeat, and Z_DNA_Motif, were predicted by non-B-gfa¹³² with the default parameters, and the total count of all types of motifs was calculated in each sequence.” (Line 955-964)

Comment 8. Within discussion, when the authors talk about the scope of their sequencing approach in terms of generating genome assemblies, the number of reads and genome size are mentioned as key factors. Nevertheless, as a highly contiguous genome assembly is always one of the goals, the amount of repetitive DNA must be considered. In other words, even in the case of a genome size and sequence coverage as those shown by the authors for *D. melanogaster*, the level of contiguity could be lower than the one shown for their genomes in Table 1 if the number of repetitive sequences was substantially higher than that in *D. melanogaster*. Definitely, long sequencing reads can help bypass the “fragmentation” impact associated with repetitive sequences but it is not totally clear to me to what extent.

Again, we appreciate this insightful comment and we are sorry for the lack of logical stringency. We updated the discussion to acknowledge the importance of TE content.

“Therefore, it is feasible to assemble genomes as large as 500 Mb, especially those with low TE or repeat content, akin to *D. melanogaster*.” (Line 686-688)

Comments of Reviewer #2 (Remarks to the Author):

The authors of the manuscript, titled "Low-Input long-read sequencing generates high-quality individual fly genomes and characterizes mutational processes", introduced LILAP, a novel library generation method specifically designed for PacBio technology. LILAP utilizes Tn5 transposase mediators and hairpin adapters to construct a circular PacBio sequencing library. In comparison to the amplification-based PacBio protocol, LILAP stands out due to its low-input requirement (100 ng), cost-effectiveness, and without amplification. In this study, the authors applied the LILAP library preparation method to sequence the genomes of two male *Drosophila melanogaster* individuals. Through a comprehensive analysis of the genomes and variants of these individuals, the authors demonstrated the efficacy of LILAP in investigating genome mutations, particularly in the context of small organisms and precious samples.

Many thanks for your comprehensive summary.

Here is a list of questions and concerns in no particular order and I hope these help improve the manuscript overall.

We deeply appreciate your kindness and constructive comments.

Comment 1. Since LILAP in this manuscript is specifically designed for PacBio technology, the title could be more specific, unless the authors can demonstrate the extension of LILAP to Oxford Nanopore technology.

Thanks. We have changed the title to "Low-input PacBio sequencing generates high-quality individual fly genomes and characterizes mutational processes".

Comment 2. Genome assembly performance is affected by sequencing depths. Thus, in Table 1, I suggest the authors generate similar depths of PacBio sequencing reads with the standard amplification-based PacBio protocol to perform a more meaningful comparison.

Thanks for this critical comment. In our previous comparisons across various studies, we overlooked controlling for confounding technical factors, such as sequencing

depth and sequencing platform. We have now incorporated two sets of amplification data generated from a single fly, ensuring a comparable depth or platform to ISO1-1/2. Once more, our ISO1-1/2 assemblies exhibited superior performance to the two assemblies based on amplification data across most evaluation parameters. As a result, we have updated Fig. 2c-d, Extended Data Fig. 4, as well as Table 1 accordingly.

The new version of the manuscript reads as:

Results:

“The remarkable coverage evenness likely stems from the amplification-free nature of LILAP or the absence of amplification bias²¹. To investigate this, we compiled two sets of individual fly sequencing data using standard amplification protocols on the PacBio Sequel II platform: one produced by Annoroad's commercial service (referred to as aISO1-Anno), and the other by PacBio itself (referred to as aISO1-PB, Materials and methods). As anticipated, both aISO1-Anno and aISO1-PB show more pronounced fluctuations (Fig. 2c), with low- or high-GC regions being underrepresented, displaying depths 10~80% lower than expected (Fig. 2d).” (Line 218-225)

“

Fig. 2 | Evaluation of LILAP sequencing data.

c, The chromosomal distribution of relative read depth with 100-kb window size. Annoroad's amplification-based sequencing data were denoted as aISO1-Anno, whereas PacBio's data were labeled as aISO1-PB. Except for the aISO1-PB, all other datasets were derived from male flies. Consequently, the presence of a minority of reads mapped to chrY in aISO1-PB likely indicates mismapping. Notably, the dots

positioned between chr2L (Left) and chr2R (Right), between chr3L and chr3R, and within chrX denote centromeres. **d**, The relative read depth in regions with different GC content. Standard PacBio HiFi sequencing data, Tn5 tagmentation-based short-read (Illumina) data, and amplification-based PacBio HiFi sequencing data served as controls. The left and right Y-axes show the relative depth and the proportion of the corresponding GC bins, respectively. The relative depth was calculated as $\frac{\text{depth of each window}}{\text{median (depth of all windows)}}$. ” (Line 183-200)

“Similarly, we conducted assemblies for two individual fly genomes using standard amplification protocols: aISO1-Anno and aISO1-PB. Subsequently, we assessed our final ISO1-1/2 assemblies against the newly generated aISO1-Anno/PB assemblies, alongside five previously published long-read fly assemblies. These include two individual fly assemblies^{9,10} and three assemblies generated by sequencing dozens or hundreds of flies^{38,43,44}. This ensemble of seven assemblies encompasses variations in sequencing platforms and depths, thereby providing a comprehensive control for evaluation.”(Line 244-251)

“The reference genome coverage exhibited even more pronounced disparities, with the ISO1-1/2 assemblies achieving 99.6% coverage for all major chromosomes, excluding the poorly assembled chrY⁴⁷⁻⁴⁹, whereas coverage for other assemblies varied between 80.9% and 99.2%.”(Line 262-266)

“Table 1. Evaluation of ISO1-1/2 assemblies and five previous long-read fly assemblies.

Study ^a	Platform ^b	Sequencing depth	DNA input ^c	Sex ^d	QV ^e	Genome coverage ^f	Gene coverage ^g	Size (Mb)	NG50 (Mb)	Median LG90 ^h
ISO1-1	PacBio	59x			60.6/64.6	99.6%	98.7%	164.1	6.7	5
ISO1-2	Sequel II HiFi mode	57x	100 ng	M	58.5/63.1	99.6%	98.7%	167.5	6.7	4
aISO1-Anno	PacBio	60x	40 ng	M	54.7	98.2%	97.9%	156.5	1.2	20
aISO1-PB	Sequel II HiFi mode	60x	10 ng	F	56.3	97.9%	98.7%	156.4	9.7	4
Berlin et al. ⁴³	PacBio Sequel II CLR mode	90x	10 µg	M	42.9	99.2%	98.3%	164.1	13.6	2

	ONT+									
Adams et al. ⁹	Illumina+	60x	78 ng	F	33.6	80.9%	92.6%	111.0	21.7	NA
	Hi-C									
Heavens et al. ¹⁰	ONT R9.4	36x	110 ng	Un	37.7	91.7%	94.4%	129.8	0.6	86
Solares et al. ⁴⁴	ONT R9.5	30x	1.5 µg	Mixed	28.5	82.8%	97.2%	140.7	2.9	11
	PacBio									
Nurk et al. ³⁸	Sequel II	40x	>1 µg	F	63.3	97.2%	98.8%	147.3	20.2	1
	HiFi mode									

- Adams *et al.* used a hybrid individual derived from the ISO1 and I38 strains⁹, while aISO1-PB and Nurk *et al.* used a hybrid population of ISO1 and A4³⁸. Other studies used the ISO1 strain exclusively. The assembled genome from Heavens *et al.* was not publicly available¹⁰, and we performed assembly independently (Materials and methods). In contrast, the other four studies made their assemblies public, and the statistics pertain to those assemblies.
- Adams *et al.* generated multiple sequencing data types from both the ONT and Illumina platforms⁹ and sequencing depth and DNA input values refer to the ONT data. Note that the ONT Flow Cell information is not available.
- The study by Nurk *et al.* lacks specific DNA quantity information³⁸. Nevertheless, considering the use of pooled flies, it is reasonable to infer that the DNA input exceeded 1 µg. DNA input for Adams *et al.* specifically refers to the ONT sequencing.
- “M”, “F” and “Un” refer to male, female, and unknown sexes, for which Heavens *et al.* did not provide specific information¹⁰.
- For ISO1-1/2 assemblies, QV values for both the initial and polished assemblies are presented, while for other assemblies, no further polishing was performed.
- To account for chromosomal differences between sexes, coverage was calculated across all chromosomes except chrY.
- Gene coverage was quantified using BUSCO complete gene models (Materials and methods).
- “Median LG90” was exclusively computed for chr2L, chr2R, chr3L, chr3R, and chrX. “NA” denotes the inability of the assembly from Adams *et al.* to cover 90% of the reference genome across the five arms.

The inclusion of two amplification-based assemblies (aISO1-Anno/PB) in our evaluation underscores the advantage of amplification-free practices. Despite utilizing the same sequencing platform (Sequel II HiFi) and similar sequencing depth (60x), both aISO1-Anno and aISO1-PB demonstrate comparatively lower genome coverage (98.2 and 97.9%, Table 1), potentially attributable to amplification bias (Fig. 2d). Consistent with this observation, they also exhibit relatively lower QV (55 and 56), likely stemming from amplification errors caused by PCR⁵⁰. Even with an increase in sequencing depth (from 60x to ~185x), these two metrics for both amplification-based assemblies exhibit only moderate changes (Supplementary Table 3), indicating that a sequencing depth of 60x suffices and that amplification bias or errors are not effectively mitigated by further depth augmentation.”(Line 275-312)

“This results from the lack of an independent scaffolding method (*e.g.*, Hi-C⁹) and a short insert size (4.5 kb vs. 8.2 kb or higher, Fig.2a and Extended Data Fig. 4). However, even for this parameter, ISO1-1/2 showed a value of 6.7 Mb (Table 1), which is better than aISO1-Anno (1.2 Mb) and the two ONT-based assemblies (0.6–2.9 Mb). Consistently, their median LG90s (number of contigs covering over 90% chromosomes) are only 4 or 5 (Table 1 and Extended Data Fig. 5), showcasing a reasonable level of continuity.

Collectively, the results indicated that the ISO1-1/2 assemblies largely outperformed other long-read assemblies from single or pooled flies, achieving a base error of 10^{-6} and near-complete coverage of reference genomes or genes.”(Line 329-338)

Extended Data Figure:

Extended Data Fig. 4 | The length distribution of CCS reads for amplification-based PacBio sequencing data.

The figure convention follows Fig. 2a.”

Comment 3. How many novel sequences do ISO1-1 and ISO1-2 have compared to the reference genome and other assemblies in this study? What is the N50 of these novel sequences?

We first confirmed that SVs identified through alignments between contigs and the reference genome (Fig. 3a) exclusively represent structural alterations of known sequences in the reference. Consequently, the 94 contigs that failed to be taxonomically annotated by searching known sequences potentially represent novel sequences. After trying different sequence search algorithms, we ultimately confirmed that 13 contigs could not be aligned back to the reference genome, suggesting that they represent novel sequences. In total, these sequences constitute 181 kb, with an N50 of 27 kb. They primarily consist of repeat-rich regions, making them challenging to sequence, assemble, and search for.

The new version of the manuscript reads as:

Results:

“Regarding the 94 contigs lacking taxonomic annotation, they likely correspond to unassembled Y-linked sequences of flies or the variable number of tandem repeats (VNTRs; Supplementary Table 7, Materials and methods). Upon further scrutinizing the reference genome, we verified that 10 potential Y-linked sequences and three VNTRs could not be aligned back to the reference genome, indicating that they represent novel sequences. In total, these sequences amount to 181 kb, with an N50 of 27 kb (Supplementary Table 7).” (Line 610-616)

Materials and methods:

“To perform taxonomic annotation and identify other potential prokaryotic content, we followed the ref.¹³⁹ and used DIAMOND v2.0.13¹⁴⁰ together with NCBI BLAST v2.12.0 to search the contigs against UniProt proteins retrieved in April 2021¹⁴¹ as well as the nucleotide collection (nt) database retrieved in February 2022¹⁴². With Blobtools2 v2.6.5¹⁴³, the results were processed, and bubble plots were generated.

Some of the 94 taxonomically unannotated contigs in ISO1-1/2 potentially belonged to unassembled Y-chromosome sequences. To identify Y-linked contigs, we separated multiplex family sequencing data into male and female groups and aligned these CCS reads to ISO1-1/2 contigs by using minimap2. The relative depth of each ISO1-1/2 contig was calculated as the median depth of the contig divided by the median depth of all contigs. Contigs meeting the criteria of having roughly half the relative depth (in comparison to known autosomal contigs) within the male group and a depth of zero in the female group were classified as Y-linked contigs.

Through manual curation, we identified the remaining 84 contigs as VNTRs. We utilized TRF v4.09¹³⁴ to generate consensus repeat units and observed that these VNTRs consist of hundreds to thousands of copies. This accounts for the failure of DIAMOND and BLAST to identify matches. Using BLAT, we successfully mapped 81 VNTRs back to the reference genome, while the remaining three VNTRs may not have been assembled in the reference.”(Line 1045-1063)

A partial snapshot of Supplementary Table 7 was shown here:

“

Individual	Contig	Length (bp)	Annotation
	ptg0000291	9021	chrUn_DS485270v1:1-1183 (CAAGA repeat)
	ptg0000541	20324	chrUn_DS485270v1:1-1183 (CAAGA repeat)
	ptg0000601	9937	chr4:797426-798097 (TATTATAT repeat)
	ptg0000791	85189	Y_linked
	ptg0001221	5007	chrUn_DS485255v1:557-968 (TAACATAGAA repeat)
	ptg0001311	10034	chrUn_DS485255v1:557-968 (TAACATAGAA repeat)
	ptg0001341	2838	chrUn_DS485255v1:557-968 (TAACATAGAA repeat)
	ptg0002051	27153	Y_linked
	ptg0002261	19004	chr2L:20018077-20018361 (CTAT repeat)
	ptg0002291	6458	chrUn_DS483998v1:48-3313 (TATTC repeat)
	ptg0002401	16321	chrUn_CP007088v1:1-366 (TGTCTT repeat)
	ptg0002431	22068	chrUn_DS484007v1:1-647 (CAAGA repeat)
	ptg0002641	20382	chrUn_CP007088v1:1-366 (TGTCTT repeat)
	ptg0002871	6765	chrUn_DS485255v1:557-968 (AACATAGAA repeat)
	ptg0002931	9892	chrUn_CP007101v1:422-788 (ATATTTT repeat)
	ptg0002941	3883	NA (GTTTTAT repeat)
	ptg0002951	8379	chrUn_DS485270v1:1-1183 (CAAGA repeat)
	ptg0003141	23184	chrUn_DS485270v1:1-1183 (ACAAG repeat)
ISO1-1	ptg0003321	15076	chr3R:5145554-5145578 (TATATGTTTTA repeat)

Supplementary Table 7. Contigs without taxonomic annotation.

For contigs lacking taxonomic annotation, “Y-linked” indicates that the contig is inferred to be a chrY fragment (Materials and methods). The remaining ones

represent VNTRs with their potential locations in the reference genome and the annotation of consensus units. It is noteworthy that three VNTRs could not be mapped back to the reference genome.”

Comment 4. In the SV analysis part, the SVs should be further validated by the standard amplification-based PacBio protocol to exclude the possibility that these mutations are introduced by the LILAP library preparation method.

We appreciate this important comment. Since the DNA of ISO1-1/2 flies has been already used, we could only perform validation with the aforementioned amplification-based aISO1-Anno sequencing data, which was derived from the same lab strain as ISO1-1/2. We expect that mutations shared by ISO1-1 and ISO1-2 should be largely shared by aISO1-Anno since the majority of them were present in the previous separate family sequencing data (Figs. 3b and 4b). Consistently, 97.5% of shared SV and 84.4% of shared SNPs are validated in aISO1-Anno. We accordingly updated the Results, Supplementary files, and Materials and methods.

The new version of the manuscript reads as:

Results:

“To eliminate the likelihood that observed SVs are artifacts from Tn5-mediated library preparation, we manually assessed 80 randomly selected shared SVs for their presence in aISO1-Anno, which was derived from the identical laboratory strain. Consistently, most SVs (78 or 97.5%) were confirmed present, while the remaining two could be explained by between-individual differences (Supplementary Table 5).”(Line 352-357)

“Notably, all nine cases were once again confirmed in aISO1-Anno (Supplementary Table 5).”(Line 405-406)

“The patterns observed for SVs (Fig. 3a-b and Supplementary Table 5) are mirrored in SNPs: 1) the majority (1,682 or 95.8%) of these SNPs were shared by both individuals and were more likely to be homozygous and covered by the two families (Figs. 4a-b and 2e); and 2) most (1,420 or 84.4%) of these SNPs were again validated in aISO1-Anno (Extended Data Fig. 11). Hence, this SNP dataset also exhibits high quality.”(Line 477-482)

Materials and methods:

“We confirmed that these SVs were not artifacts induced by LILAP. Given the extensive manual curation efforts in SV identification, validation was carried out only on a subset of SVs utilizing amplification-based sequencing data from aISO1-Anno. A total of 89 SVs were chosen, including 80 randomly selected SVs (simple transpositions or other types of SVs) shared by ISO1-1 and ISO-2, along with nine complex transpositions. Following the extraction of SV sequences from ISO1-1, they were mapped to the aISO1-Anno assembly, and CCS reads overlapping the focal sites were extracted. These reads were then aligned to the reference genome using BLAT, with subsequent manual curation conducted in the UCSC Genome Browser.”(Line 940-948)

“The identical workflow, encompassing PAV and subsequent filters utilized for ISO1-1/2 assemblies, was applied to the aISO1-Anno assembly. An in-house Python script was utilized to automatically identify overlaps between SNPs in aISO1-Anno and ISO1-1/2, based on the consistency of their positions and mutation directions.”(Line 991-994)

Extended Data Figure:

Extended Data Fig. 11 | SNP distribution across different ISO1 individuals

The figure convention follows Extended Data Fig. 7a.”

A partial snapshot of Supplementary Table 5 was shown here:

“

REF_chr	Start	End	Description	Type	Simple/complex	ISO1-1_contig	ISO1-1_start	ISO1-1_end	Status in aISO1- Anno
chr2L	6594317	6594547	DNA transposon insertion with 229-bp deletion	Insertion	Complex	ptg000010l	190792	192461	Present
chr2L	8741277	8741281	LINE retrotransposon insertion with CAA deletion	Insertion	Complex	ptg000026l	8420997	8425675	Present
chr2L	9404259	9404259	DNA transposon insertion and FBgn0027600 partial duplication	Insertion	Complex	ptg000026l	7731298	7727749	Present
chr2R	13586682	13586863	LINE retrotransposon insertion with 180-bp deletion	Insertion	Complex	ptg000016l	3414355	3419061	Present
chr3R	10541217	10541241	DNA transposon insertion with 23-bp deletion and 45-bp duplication	Insertion	Complex	ptg000015l	6999554	7001006	Present
chr3R	22469141	22469341	DNA transposon insertion with 199-bp deletion	Insertion	Complex	ptg000006l	3396028	3397434	Present
chr3R	24684679	24684882	DNA transposon insertion with 211-bp deletion	Insertion	Complex	ptg000006l	1165644	1167051	Present
chr3R	29776816	29777196	LINE retrotransposon insertion with 379-bp deletion	Insertion	Complex	ptg000012l	57699	62402	Present
chrX	15816858	15816881	DNA transposon insertion with 22-bp deletion	Insertion	Complex	ptg000035l	1001062	1004044	Present
chr2L	4366791	4366801	LINE retrotransposon insertion	Insertion	Simple	ptg000010l	2432460	2432472	Present
chr2L	6471813	6471826	LINE retrotransposon insertion	Insertion	Simple	ptg000010l	317784	317796	Present

Supplementary Table 5. SV validation in aISO1-Anno.

The table presents nine complex transpositions as well as 80 randomly selected ISO1-1/2 shared SVs. The "Description" column provides details of each SV. The "Simple/Complex" column distinguishes nine complex transpositions. The "State in aISO1-Anno" column indicates whether the SV has been confirmed present in the aISO1 annotation ("Present") or *vice versa* ("Absent")."

Comment 5. What does the y-axis of the first panel in Fig 1b represent? Insects?

Thank you for pointing out this issue. We have clarified it as "Species number".

Fig. 1 | Summary of low-input library preparation methods, project design, and schema of LILAP.

b, The overall design of this study. A log-scaled axis was used for the leftmost panel. Cartoons for three exemplar insects are shown with *Kikiki huna* in the left, *D. melanogaster* in the middle, and *Coccinella septempunctata* in the right. For PacBio sequencing, the red point indicates polymerase and the yellow line marks the transposase binding sequence (see also Panel c). CCS: circular consensus sequencing; SV: structural variation; SNP: single nucleotide polymorphism." (Line 118-131)

Comment 6. In Fig. 2c, could the centromere region be labelled?

Sorry for this negligence. We have now labeled the centromeres.

“

Fig. 2 | Evaluation of LILAP sequencing data.

c,...Notably, the dots positioned between chr2L (Left) and chr2R (Right), between chr3L and chr3R, and within chrX denote centromeres.” (Line 183-195)

Comments of Reviewer #3 (Remarks to the Author):

In this study, a low-input, low-cost, and amplification-free library preparation method is proposed for PacBio-based genome sequencing. The method is evaluated on two *Drosophila melanogaster* individuals and demonstrates multiple promising features, including high genome coverage, low sequencing errors (0.02%), and the potential to resolve structural variations and symbiont genomes. Overall, the method is impressive and has a wide range of applications. Before acceptance for publication, I have three minor questions.

Many thanks for these encouraging comments.

Comment 1. The authors discuss the potential of individual-level GWAS with such a low-cost library preparation method, which is quite promising. My question is why the authors do not demonstrate a demo example of this interesting concept, given the advantages of this method. More details should be added or discussed on the promises and difficulties in implementing individual-level GWAS analysis.

Thank you for the insightful comment. The reason we did not conduct individual-level GWAS analysis is due to the small number of flies and the scope of the manuscript. We have now revised Fig. 6 and provided a more detailed discussion on individual-level GWAS analysis.

The new version of the manuscript reads as:

Discussion:

“Second, conventional genome-wide association studies (GWAS) of flies are based on short-read population sequencing⁹⁵, which can be confounded by incomplete genetic information and genetic diversity between individuals (Figs. 3a and 4a)^{96,97}. Individual genomes generated by LILAP effectively address this issue (Fig. 6b), although individual-level GWAS analysis was not conducted here due to the limited number of single flies and the extensive scope of topics covered in this study. Coupled with rapidly evolving high-throughput phenotyping techniques (e.g., automatic imaging analyses⁹⁸), individual-level GWAS holds promise for efficiently unraveling the genotype-phenotype map.” (line 688-696)

“

Fig. 6 | Potential application directions of LILAP.

a, LILAP for biodiversity exploration and somatic mutations. The top and middle panels depict the harvesting of DNA for sequencing from a whole small insect and a part of a large insect, respectively. The bottom panel illustrates the analysis of somatic mutations specific to certain tissues. **b**, LILAP for individual-level GWAS analysis. The top, middle, and bottom panels demonstrate three stages of GWAS, including sampling, individual-level genotyping and phenotyping, and identification of association signals, respectively. **c**, LILAP for host-symbiont survey. The top and middle panels illustrate midgut symbionts and parasites, respectively, while the bottom panel shows circular and linear genomes of the microorganisms.”(Line 653-663)

Comment 2. The innovation of LILAP was stimulated by the need for insect studies. However, LILAP can be applied to more types of studies beyond insects. Therefore, the authors should show an example or add a discussion section to detail the potential applications (and potential limitations) outside of insect studies, given the ability of LILAP to derive both genome assembly and variations (0.02% error rate).

Many thanks for this important comment. We fully agree that LILAP extends beyond insect studies. We have now modified Fig. 6 and discussed one potential application, not necessarily related to insects.

The new version of the manuscript reads as:

Discussion:

“Fourth, many organisms, such as large insects or vertebrates, may yield more than 100 ng of DNA. For such cases, LILAP could exclusively utilize DNA extracted from specific body parts, such as wings or skin, to conserve materials (Fig. 6a, the middle and bottom panels). In this regard, LILAP could be also employed to investigate genomic differences within a single body, including somatic transpositions¹⁰¹ or other types of SVs in both normal and tumor contexts¹⁰².”(Line 701-707)

Comment 3. Genome assembly and annotation are currently still challenging. It is of great merit that the authors also share their code in Git Hub. A discussion section on how LILAP facilitates or imposes difficulties on genome assembly or annotation should be added. If an automatic workflow is available, it will shift the paradigm of genotype-phenotype studies.

We appreciate this insightful comment. Previously, we shared all our codes on GitHub. Now, we have also developed an automatic workflow. Additionally, in the Discussion section, we have compared LILAP with conventional PacBio sequencing and amplification-based PacBio sequencing regarding the computational framework for assembly and mutation annotation.

The new version of the manuscript reads as:

Discussion:

“An easily accessible computational framework for genome assembly and mutation annotation would enhance the widespread utilization of LILAP across all four aforementioned avenues. Consequently, we have made our codes publicly available, with the majority of them integrated into a single automated pipeline (see also Code availability). Except for the Tn5 binding sites filtering code, the remaining codes can also be applied to conventional PacBio sequencing data, which shares substantial similarities with LILAP data. Likewise, they can be adapted for amplification-based data, although additional codes are necessary to optimize the assembly of underrepresented genomic regions (Fig. 2d) and to eliminate artificial SVs caused by template switching during amplification²².”(Line 708-717)

“Code availability

The codes produced in this study are available at <https://github.com/Zhanglab-IOZ/LILAP>. The codes for genome assembly, evaluation, polishing, and SV detection have been encapsulated as an integrated Snakemake v8.0.1¹⁴⁴ workflow. To facilitate user testing, demo data has also been provided. Users have the flexibility to execute the workflow as a whole or to run individual components as needed, given that both the Snakemake workflow and the raw codes are available. It is worth noting that codes for SNP calling or *Wolbachia* analyses were not integrated into the Snakemake workflow due to technical constraints (e.g., interaction with external data sources). Instead, the raw codes have been released on GitHub as well.”(Line 1083-1093)

Supporting references

1. Berlin, K. *et al.* Assembling large genomes with single-molecule sequencing and locality-sensitive hashing (vol 33, pg 623, 2015). *Nature Biotechnology* **33**, 1109-1109 (2015).
2. Rahman, R. *et al.* Unique transposon landscapes are pervasive across *Drosophila melanogaster* genomes. *Nucleic Acids Res* **43**, 10655-72 (2015).
3. Chakraborty, M., Emerson, J.J., Macdonald, S.J. & Long, A.D. Structural variants exhibit widespread allelic heterogeneity and shape variation in complex traits. *Nat Commun* **10**, 4872 (2019).

REVIEWERS' COMMENTS

Reviewer #1 (Remarks to the Author):

I commend the authors for the work done to address my concerns. In the new version, the work done is better justified and the different components of the manuscript are better integrated.

Reviewer #2 (Remarks to the Author):

All of my concerns have been well addressed. I am happy with current revision.

Reviewer #3 (Remarks to the Author):

The authors have fully addressed all my concerns.

Reviewer #3 (Remarks on code availability):

The code has been well annotated and publicly available.